# OmniAudio: Generating Spatial Audio from 360-Degree Video

**Huadai Liu** [1 2]  **Tianyi Luo** [3]  **Kaicheng Luo** [3]  **Qikai Jiang** [3]  **Peiwen Sun** [1]  **Jialei Wang** [3]  **Rongjie Huang** [4]
**Qian Chen** [2]  **Wen Wang** [2]  **Xiangtai Li** [5]  **Shiliang Zhang** [2]  **Zhijie Yan** [2]  **Zhou Zhao** [3]  **Wei Xue** [1]

## Abstract

Traditional video-to-audio generation techniques primarily focus on perspective video and non-spatial audio, often missing the spatial cues necessary for accurately representing sound sources in 3D environments. To address this limitation, we introduce a novel task, **360V2SA**, to generate spatial audio from 360-degree videos, specifically producing First-order Ambisonics (FOA) audio - a standard format for representing 3D spatial audio that captures sound directionality and enables realistic 3D audio reproduction. We first create **Sphere360**, a novel dataset tailored for this task that is curated from real-world data. We also design an efficient semi-automated pipeline for collecting and cleaning paired video-audio data. To generate spatial audio from 360-degree video, we propose a novel framework **OmniAudio**, which leverages self-supervised pre-training using both spatial audio data (in FOA format) and large-scale non-spatial data. Furthermore, OmniAudio features a dual-branch framework that utilizes both panoramic and perspective video inputs to capture comprehensive local and global information from 360-degree videos. Experimental results demonstrate that OmniAudio achieves state-of-the-art performance across both objective and subjective metrics on Sphere360. Code and datasets are available at `github.com/liuhuadai/OmniAudio`. The project website is available at `OmniAudio-360V2SA.github.io`.

## 1. Introduction

The rapid advancement of virtual reality and immersive technologies has created an urgent need for realistic audio-visual experiences. Although many video-to-audio generation methods (Iashin & Rahtu, 2021; Luo et al., 2024; Zhang et al., 2024; Mei et al., 2024) are proposed, they face two critical limitations. First, they typically generate non-spatial audio (mono/stereo), which lacks the directional information essential for immersive experiences. Second, they operate on perspective videos with limited field-of-view, missing crucial visual context for sound generation, as illustrated in Figure 1.

Spatial audio, particularly in First-order Ambisonics (FOA) format (Zotter & Frank, 2019), offers a solution to the first limitation by preserving 3D sound positioning. Recent works have begun exploring spatial audio generation: Anonymous (2024) generate FOA audio from video perspectives using autoregressive models, Heydari et al. (2024) synthesize FOA audio from text and room parameters, and Kushwaha et al. (2024) condition on sound categories and source locations. However, these approaches still rely on fixed camera perspectives, inheriting the second limitation of restricted visual context.

To overcome both limitations, we introduce 360-degree video to spatial-audio generation (**360V2SA**), a novel task that generates FOA audio directly from 360-degree (panoramic) videos. This task is particularly timely given the increasing accessibility of 360-degree cameras. Panoramic videos offer significant advantages over traditional perspective videos - they capture a complete 360-degree field of view, allowing simultaneous observation of all sound-emitting objects and their spatial relationships, regardless of direction. By utilizing this comprehensive spherical visual coverage, 360V2SA enables the the generation of spatially-aware audio that naturally aligns with the visual content, without requiring additional control parameters like camera angles.

The 360V2SA task presents three major challenges: (1) the scarcity of paired 360-degree video and spatial audio data, (2) the need for precise audio-visual synchronization across the entire sphere, and (3) the complexity of generating high-fidelity spatial audio.

---

[1]Hong Kong University of Science and Technology, Hong Kong, China [2]Tongyi Lab, Alibaba Group, Hangzhou, China [3]Zhejiang University, Hangzhou, China [4]FAIR, Meta, USA [5]Nanyang Technological University, Singapore. Correspondence to: Wei Xue <weixue@ust.hk>, Zhou Zhao <zhaozhou@zju.edu.cn>.

*Proceedings of the $42^{nd}$ International Conference on Machine Learning*, Vancouver, Canada. PMLR 267, 2025. Copyright 2025 by the author(s).

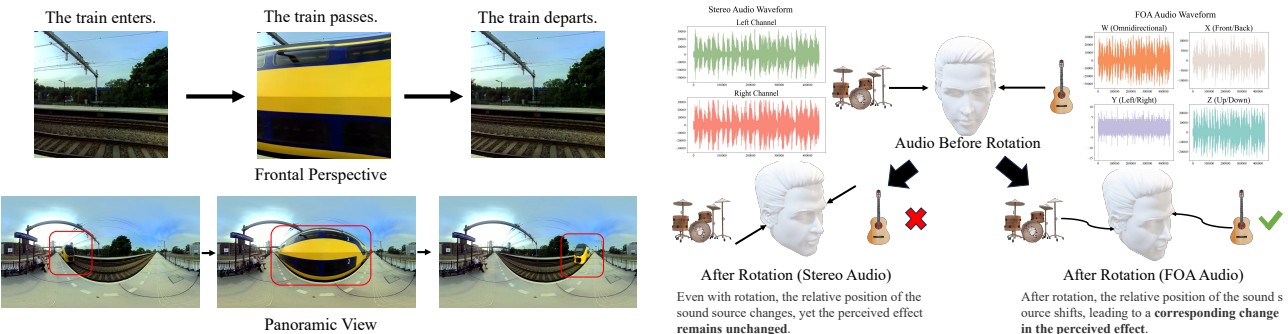

(a) Comparison of panoramic video and perspective video.    (b) Comparison of stereo audio and FOA audio under head rotation.

*Figure 1.* (a) shows the scene of a moving train that appears and gradually disappears in a panoramic view without being visible in the frontal perspective. (b) compares the audio localization before and after head rotation, illustrating how stereo audio fails to maintain sound localization while spatial audio (in FOA format) retains accurate positioning.

We propose an end-to-end framework **OmniAudio** for 360V2SA. To address the data scarcity challenge, we construct **Sphere360**, the first large-scale dataset for 360V2SA, containing 103,000 real-world video clips, each with a 10-second duration, spanning 288 audio events. We develop a semi-automated pipeline for dataset construction. Moreover, we also propose a novel training strategy in OmniAudio that leverages existing non-spatial audio datasets through a spatial autoencoder and masked token prediction pre-training, in a self-supervised coarse-to-fine manner. By masking and reconstructing portions of audio tokens, the model learns general audio patterns that transfer to spatial audio generation via the autoencoder, effectively bridging the domain gap. For precise audio-visual synchronization across the entire sphere and generating high-fidelity spatial audio, we design a dual-branch architecture in OmniAudio that combines latent flow matching with both local and global video processing. This architecture captures fine-grained object details while maintaining awareness of the complete spherical context, enabling precise audio-visual synchronization through a two-stage optimization process.

Our contributions can be summarized as follows:

- We introduce 360V2SA, a novel task addressing fundamental limitations in traditional video-to-audio generation, and propose OmniAudio, an end-to-end solution using latent flow matching.
- We develop an effective training strategy combining coarse-to-fine pre-training and dual-branch video encoding for spatial-aware generation.
- We create and release Sphere360, a comprehensive dataset of 103,000 video clips with spatial audio, along with its semi-automated construction pipeline.
- We establish Sphere360-Bench as a standardized evaluation framework where our OmniAudio achieves state-of-the-art performance across both objective and subjective metrics.

## 2. Related Work

**Video-to-Audio Generation.** This direction (Zhang et al., 2024; Wang et al., 2024a; Xu et al., 2024c; Tian et al., 2024) focuses on synthesizing audio that aligns seamlessly with the visual content of a video clip. Some approaches, such as SpecVQGAN (Iashin & Rahtu, 2021), FoleyGen (Mei et al., 2024), and V-AURA (Viertola et al., 2024), leverage autoregressive techniques to produce audio from silent video. Meanwhile, other methods (Luo et al., 2024; Cheng et al., 2024b; Xu et al., 2024b) adopt diffusion models (Song et al., 2020; Xing et al., 2024; Liu et al., 2024a) or flow matching (Lipman et al., 2022; Vyas et al., 2023; Wang et al., 2024b) generative models. Diff-Foley (Luo et al., 2024) employs an audio-visual contrastive feature and latent diffusion to predict spectrogram latent. Similarly, MovieGen Audio (Polyak et al., 2024) uses flow matching conditioned on multi-modal inputs, including videos and texts. Despite these advancements, a significant gap remains in the research focused on generating spatial audio from 360-degree video, an essential component for crafting genuinely immersive experiences. We introduce OmniAudio, which, to the best of our knowledge, represents the **first approach** in spatial audio generation from 360-degree video.

**Spatial Audio Generation.** Existing spatial audio [1] generation methods (Xu et al., 2021; Leng et al., 2022; Liu et al., 2023c; Gao & Grauman, 2019b; Kushwaha et al., 2024; Anonymous, 2024) predominantly focus on producing binaural (Sun et al., 2024; Yoshida et al., 2023; Garg et al., 2021) or First-order Ambisonics (FOA) audio (Kushwaha et al., 2024; Dagli et al., 2024) from fixed perspective inputs, such as mono audio, visual features, text, and

---

[1] Binaural audio captures sound using two microphones to simulate human ear positioning for a 3D effect when using headphones, while spatial audio involves a broader range of techniques that create 3D sound in dynamic environments, including with head tracking or multi-speaker setups.

source locations. For example, 2.5D Visual Sound (Gao & Grauman, 2019b) transforms monoaural audio into binaural sound using convolutional neural networks, while Diff-SAGe (Kushwaha et al., 2024) and ViSAGe (Anonymous, 2024) generate FOA audio by incorporating sound categories, source locations, and camera parameters. However, these approaches typically assume a static camera perspective, limiting their applicability in dynamic environments where the field of view can change in real-time. To overcome these limitations, we propose a framework for generating FOA audio from 360-degree video, which inherently supports dynamic and panoramic visual inputs. Our method leverages self-supervised flow matching pre-training and dual-branch video design to effectively synthesize spatial audio that adapts to the comprehensive and evolving visual context captured by 360-degree videos. This advancement enables more immersive and flexible audio-visual experiences in applications requiring real-time spatial audio adjustments.

**Flow Matching.** Flow matching is used as the backbone for audio generation due to its superior generation performance. This framework (Lipman et al., 2022) trains a model to learn a vector field, enabling the generation of a desired probability path from noise to data. Unlike score-based models such as Denoising Diffusion Probabilistic Models (DDPM) (Ho et al., 2020), FM offers more stable and robust training and enhanced performance. This generative model has proven effective in audio generation, as seen in applications like Audiobox (Vyas et al., 2023) and FlashAudio (Liu et al., 2024b). Moreover, SpeechFlow (Liu et al., 2023a) has integrated self-supervised pre-training with FM to improve speech processing tasks, demonstrating its adaptability. In our work, we adapt FM for the novel task of spatial audio generation from 360-degree videos. By adopting a coarse-to-fine self-supervised pre-training strategy, our approach first learns general audio patterns from large-scale non-spatial audio data and then fine-tunes the model to capture spatial-specific characteristics. This enables the generation of high-quality First-order Ambisonics (FOA) audio that complements the immersive visual context of 360-degree videos.

## 3. The Sphere360 Dataset

**Overview.** The Sphere360 dataset comprises over 103,000 paired audio and 360-degree video clips sourced from YouTube, amounting to 288 hours. The dataset covers a wide range of real-world acoustic environments and noise conditions. We partition the dataset into training and test sets based on video IDs, to maintain the integrity of audio event distributions for evaluation and prevent data leakage. We implement a comprehensive data filtering pipeline to ensure high-quality video and audio samples. Please refer to Figure 8 for the distribution of different audio events.

**Data Collection.** Based on YT-360 (Morgado et al., 2020), we construct the dataset by crawling YouTube, leveraging its extensive collection of 360-degree videos and spatial audio content. The data collection process follows a systematic approach with several key steps. The initial search strategy involves using carefully formulated search keywords (e.g., "skiing spatial audio 360") to ensure class diversity and retrieve more 360-degree and FOA content. Technical filtering is applied to exclude videos that do not support either 360-degree visual content or FOA audio. Guided by target lists and technical filters, the crawling process proceeds in two stages: first, a channel-based approach where we identify frequently appearing channels and collect videos on a per-channel basis, covering the majority of search results; second, a video-based approach, which involves manually reviewing the remaining videos and applying final filters for dataset completion. Throughout the process, we maintain rigorous quality assurance through periodic re-evaluations of search results to refine keywords, as well as by sampling clips within target channels and videos to manually filter out content with unrealistic scenes or excessive post-production modifications. Detailed copyright and collection protocol information is provided in Appendix A and B.1, respectively.

**Data Cleaning.** While the initial collection focuses on gathering the 360-degree video and FOA audio, we further clean the dataset to remove stationary videos, silent audio, excessive speech, and videos with mismatched audio-visual content from the dataset. Semi-automated processes are adopted. Stationary videos are identified by measuring the similarity between frames using the mean squared error (MSE), and video segments containing more than 85% stationary frames are removed. Silent audio segments are identified using a 20ms sliding window to calculate the maximum decibels relative to full scale (dBFS), with clips containing over 90% silence being filtered out. To remove excessive speech content, we employ the SenseVoice Large model[2] for speech detection. Videos containing more than 5 detected words are filtered out, though we retain those with minimal vocal content to preserve naturally occurring sounds. Finally, we ensure strong audio-visual correspondence using Imagebind (Girdhar et al., 2023) for alignment assessment, removing videos with similarity scores below 1. Detailed cleaning procedures are documented in Appendix B.2.

**Comparison with Existing Datasets.** As shown in Table 1, Sphere360 stands out as the largest 360-degree video dataset with FOA audio format, specifically tailored for 360-degree video-to-spatial audio generation. In comparison, YT-360, previously the largest dataset providing 360-degree

---

[2] https://github.com/FunAudioLLM/SenseVoice

| Dataset | #Clips | Total Duration | V/A Type | | Audio Generation |
|---------|--------|----------------|----------|------|------------------|
| | | | 360° | FOA | |
| VGGSound (Chen et al., 2020) | 200K | 550h | ✗ | ✗ | ✓ |
| FairPlay (Gao & Grauman, 2019a) | 1.8K | 5.2h | ✗ | ✗ | ✗ |
| OAP (Vasudevan et al., 2020) | 64K | 15h | ✓ | ✗ | ✗ |
| REC-STREET (Morgado et al., 2018) | 123K | 3.5h | ✓ | ✓ | ✗ |
| YT-ALL (Morgado et al., 2018) | 3976K | 113h | ✓ | ✓ | ✗ |
| YT-Ambigen (Anonymous, 2024) | 102K | 142h | ✗ | ✓ | ✓ |
| STARRS23 (Shimada et al., 2023) | 0.2K | 7.5h | ✓ | ✓ | ✓ |
| YT-360 (Morgado et al., 2020) | 89K | 246h | ✓ | ✓ | ✗ |
| **Sphere360** | **103K** | **288h** | ✓ | ✓ | ✓ |

videos with FOA, offers a considerable volume of video content (approximately 246 hours). However, as evidenced by ViSAGe (Anonymous, 2024), YT-360 is not optimized for video-to-audio generation (*a competitive generative model on VGGSound struggles to train or finetune with YT360, often producing noise-like sounds as outputs*), probably due to the less stringent quality control. Here, Sphere360 has been developed using meticulously designed data collection and cleaning protocols, ensuring that the dataset adheres to high-quality standards, making it a reliable resource for research in 360-degree video-to-spatial audio generation. Additional related works and detailed comparative analyses are provided in Appendix C.

# 4. OmniAudio

**Overview.** As illustrated in Figure 2, OmniAudio consists of two main stages: (1) we employ a coarse-to-fine self-supervised flow matching pre-training (Figure 2a) to alleviate the issue of data scarcity using both unlabeled spatial and non-spatial audio. (2) In the fine-tuning stage (Figure 2b), we fine-tune the diffusion transformer by efficiently integrating panoramic video representation. In the following, we describe these stages in detail.

## 4.1. Preliminaries

**Conditional Flow Matching.** Conditional flow matching is an effective generative technique applied across image (Esser et al., 2024; Lipman et al., 2022), audio (Iashin & Rahtu, 2021; Liu et al., 2024b), and video domains (Liu et al., 2024c). It employs a time-dependent velocity vector field parameterized by a neural network, denoted as $v_\theta(t, C, x)$, where $t$ represents time, $C$ is the conditioning input (e.g., video or text), and $x$ is a data point in $\mathbb{R}^d$.

The model is trained by minimizing an objective function aimed at optimizing $\theta$. Specifically, linear interpolation is defined as:

$$x_t = tx_1 + (1 - t)x_0, \tag{1}$$

where $x_1$ are data points from the training distribution conditioned on $C$, and $x_0$ is sampled from the standard normal distribution. The velocity field $u(x_t \mid x_0, x_1)$ at any intermediate point $x_t$ is given by:

$$u(x_t \mid x_0, x_1) = x_1 - x_0. \tag{2}$$

The objective for conditional flow matching is:

$$\mathbb{E}_{t,q(x_0),q(x_1,C)} \left[ \|v_\theta(t, C, x_t) - u(x_t \mid x_0, x_1)\|^2 \right], \tag{3}$$

where the expectation is over $q(x_0)$ (noise), $q(x_1, C)$ (training data), and $t$ uniformly sampled from $[0, 1]$.

## 4.2. Spatial Audio Representation

Variational Autoencoder (VAE) plays an essential role in audio processing by compressing audio signals into latent representations (Hsu et al., 2017; Caillon & Esling, 2021; Evans et al., 2024). Traditionally, audio VAEs have involved converting waveforms into mel-spectrograms before compressing them into latent spaces. However, recent advancements demonstrate that the incorporation of Snake activations in the Descript Audio Codec architecture significantly enhances audio reconstruction quality at high compression ratios (Kumar et al., 2023), outperforming alternatives like EnCodec (Défossez et al., 2022). Building on these developments, the Stable Audio framework (Evans et al., 2024) succeeded in directly compressing stereo waveforms into latent representations, marking an advance in audio VAEs. Despite advances, there is still a lack of VAEs for four-channel FOA audio. The channels W, X, Y, and Z serve different roles: W captures overall sound pressure, X differentiates front and back sounds, Y differentiates left and right, and Z encodes vertical audio.

To address this gap and effectively encode FOA audio, we propose several modifications based on the Stable Audio framework: (1) We initialize our four-channel VAE with weights from a pre-trained stereo VAE to leverage existing non-spatial audio knowledge. (2) We eliminate the Mid-Side Short Time Fourier Transform (MS-STFT), which is specifically designed for stereo reconstruction, and adapt the system to transform the left/right components into FOA components W, X, Y, and Z, each with an loss weight of 1/4. For detailed information on VAE, please refer to Appendix D.

## 4.3. Dual-Branch Video Representation

We propose a dual-branch architecture to effectively model spatial-temporal dynamics in 360-degree videos. Given an

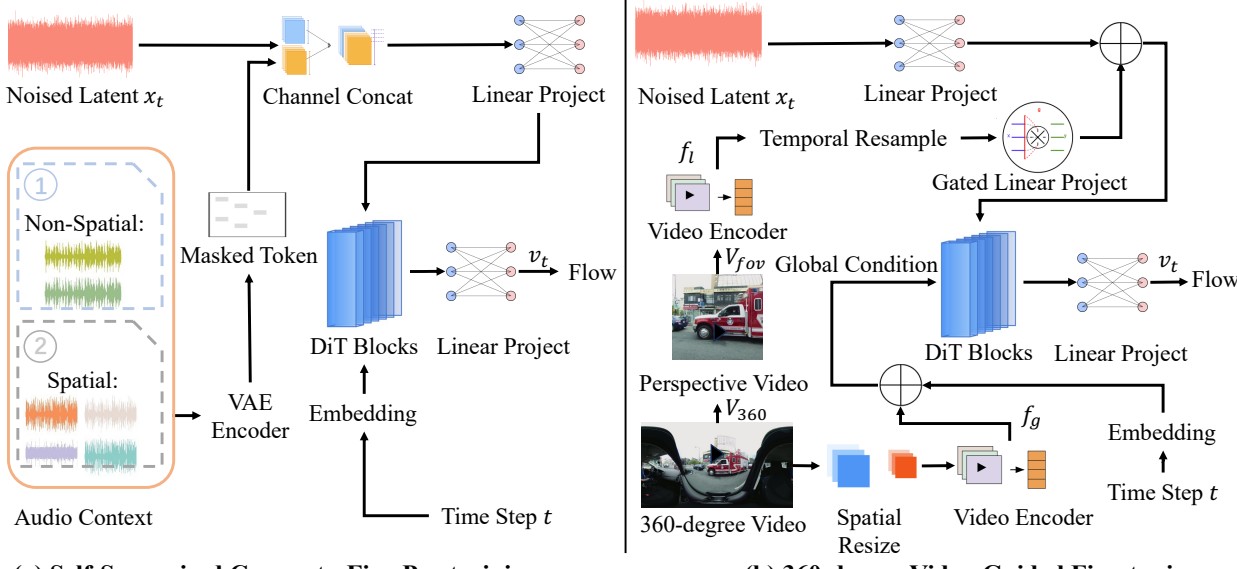

**(a) Self-Supervised Coarse-to-Fine Pre-training**

**(b) 360-degree Video-Guided Fine-tuning**

*Figure 2.* A high-level overview of OmniAudio. The model leverages stereo and FOA audios for self-supervised pre-training using token masking. OmniAudio efficiently trains for conditional generation during fine-tuning, supported by robust panoramic video representation. DiT denotes Diffusion Transformer.

input video sequence $V_{360} \in \mathbb{R}^{T \times H \times W \times C}$, which is in equirectangular projection format [3] with a height-to-width ratio of 1:2, we first pad it to a square format (1:1) and then resize it to match the input requirements of the image encoder. Here, $T$ is the number of frames, $H$ is the frame height, $W$ is the frame width, and $C$ is the number of color channels. We extract the perspective video [4], denoted as $V_{\text{fov}} \in \mathbb{R}^{T \times H' \times W' \times C}$, from the 360-degree video $V_{360}$ to capture local information. Our framework employs two complementary branches leveraging a frozen pre-trained MetaCLIP-Huge image encoder ([Xu et al., 2024a](#)):

$$ f_g = \mathcal{E}_{\text{metaclip}}(V_{360}), \quad f_l = \mathcal{E}_{\text{metaclip}}(\mathcal{P}(V_{fov})) $$

where $\mathcal{E}_{\text{metaclip}}$ processes both *global panoramic* features through equirectangular projection and *local perspective* features via linear projection $\mathcal{P}(\cdot)$. The dual representations are fused through diffusion transformers. This parameter-efficient design enables simultaneous modeling of both global scene context and fine-grained field-of-view details, crucial for high-fidelity FOA audio synthesis.

---

[3]The Equirectangular Projection (ERP) is a popular method for projecting 360-degree videos, commonly used by platforms like YouTube. It maps a spherical view onto a 2D image using latitude and longitude coordinates, with a 2:1 aspect ratio (360° horizontal, 180° vertical).

[4]Perspective video denotes standard rectilinear projections with less than 120-degree horizontal field-of-view (FoV).

## 4.4. Self-supervised Flow Matching Pre-training

The scarcity of pair 360-degree video-spatial audio dataset presents a significant challenge when compared to the abundance of unlabeled non-spatial audio resources. To address this issue, we adopt a novel self-supervised coarse-to-fine pre-training approach that leverages both the extensive unlabeled non-spatial audio and our curated FOA audio data. Inspired by the successful application of masked audio contexts in frameworks like Audiobox ([Vyas et al., 2023](#)) and Voicebox ([Le et al., 2023](#)), we propose a two-phase pre-training paradigm with masking strategy.

In the initial phase, we utilize the vast non-spatial audio dataset. We first convert the non-spatial audio into FOA audio and then compress these inputs into latent representations using our spatial VAE, and apply token masking to these latent representations. Specifically, we condition the velocity vector field $v_t$ on partially masked audio latents $x_{\text{mask}}$ with a probability $p_{\text{cond}}$ during training. This approach means the model has a $1 - p_{\text{cond}}$ chance of receiving a fully masked $x_{\text{mask}}$. The masked condition $x_{\text{mask}}$ is obtained by randomly selecting $n_{\text{mask}}$ frames to be masked, with a minimum masking span length of $l_{\text{mask}}$. This phase allows the model to learn general audio characteristics and temporal structures in a resource-efficient manner.

In the subsequent phase, we focus exclusively on FOA audio as input. This refined stage serves to pre-train the model to the specific characteristics and spatial dynamics of FOA data. By narrowing the focus to only FOA audio in the latter phase, the model effectively aligns its learned representation

to these more complex auditory patterns, thus enhancing its capability for high-fidelity FOA audio generation from 360-degree video content. We use Equation 3 for our training objective for only masked pieces.

### 4.5. Spatial-Aware Supervised Fine-tuning

Building upon our pre-trained flow matching model and dual-branch video representation, we implement an efficient fine-tuning strategy for video-guided spatial audio generation. Given a panoramic video sequence $V_{360}$ and its corresponding perspective video $V_{FOV}$, we integrate video features with dual-branch design while the FOV features are upsampled to match the audio latent sequence length and combined through element-wise addition, while 360 features pass by a max-pooling layer and then serve as a global condition in the diffusion transformer. The conditional flow matching during fine-tuning is formulated as:

$$\mathbb{E}_{t,q(x_0),q(x_1,V_{360})}\left[\|v_\theta(t, f_{360}, f_{fov}, x_t) - u(x_t \mid x_0, x_1)\|^2\right],$$
(4)

where the time steps $t$ are sampled from a logit-normal distribution. During inference, we sample trajectories using the learned velocity field with the dual-branch conditioning strategy, followed by the VAE decoder to generate the final FOA audio output.

## 5. Experiments

### 5.1. Experimental Setup

**Datasets.** The non-spatial audio datasets we utilize include FreeSound (Fonseca et al., 2017), AudioSet (Gemmeke et al., 2017), and VGGSound (Chen et al., 2020), comprising a total of approximately 2M samples. We first segment each video into 10-second clips, then sample the video at 8 frames per second (fps), with audio sampled at 44.1 kHz. To incorporate non-spatial audio into the pre-training phase, we convert them to the FOA format. Specifically, the Y and Z channels are initialized as zero, the W channel is set as the sum of the two original audio channels, and the X channel is set as the difference between the two original audio channels. Details of constructing our Sphere360-Bench benchmark are in Appendix C.

**Evaluation Metrics.** We conduct comprehensive evaluations using both objective and subjective metrics to measure spatial audio quality and video-audio alignment. (1) Objective Evaluation: For **Non-Spatial Metrics**, we adopt the widely used **Fréchet Distance (FD)** (Kilgour et al., 2018; Copet et al., 2024) to measure the similarity between the feature distributions of generated and reference audio, leveraging the OpenL3 feature space for audio projection (Cramer et al., 2019; Evans et al., 2024). Additionally, we use the **Kullback-Leibler (KL) Divergence** to measure the difference between label distributions of generated and reference

audio (Copet et al., 2024), with the metric computed using the PaSST model trained on AudioSet (Koutini et al., 2021). For **Spatial Audio Metrics**, following Heydari et al. (2024), we employ **directions of arrival (DoA)** and the mean error to evaluate spatial audio quality. Specifically, we use three spatial angle-related metrics: $\Delta_{abs}\theta$, $\Delta_{abs}\phi$, and $\Delta_{Angular}$. (2) Subjective Evaluation: We conduct human evaluation using the **Mean Opinion Score (MOS)** to evaluate both spatial audio quality (**MOS-SQ**) and video-audio alignment faithfulness (**MOS-AF**). Details of the objective metrics and subjective evaluation can be found in Appendix E.1 and Appendix E.2.

**Baselines.** Since our work pioneers in generating spatial audio from 360-degree videos, we construct the following baselines for comparison: (1) A cascaded system that integrates the state-of-the-art (video + text)-to-audio generation model MMAudio (Cheng et al., 2024a) with an audio spatialization component. Appendix E.4 includes details of the spatialization component. (2) Another cascaded system that integrates the classic video-to-audio generation model Diff-Foley (Luo et al., 2024) with the audio spatialization component. (3) ViSAGe (Anonymous, 2024), a recent model designed for generating spatial audio from perspective video inputs. We implement both ViSAGe with FoV video [5] inputs and ViSAGe with panoramic videos for comparisons. ViSAGe is considered a competitive model as it specifically targets spatial audio generation. For all baselines except ViSAGe (360), we use the field-of-view of panoramic videos as video inputs, while for ViSAGe (360), we use the 360-degree video as input. Both baselines are trained on the Sphere360 dataset. We assess the performance of OmniAudio and baseline models on both the Sphere360-Bench and the YT360 test set. The YT360 test set is particularly valuable for evaluating the generalizability of our model due to its out-of-distribution characteristics. Note that audio spatialization is essentially the inverse process of calculating spatial angles: the spatialization is performed using the ground truth spatial angles. Therefore, "+AS" models are not evaluated using the three metrics related to evaluating spatial angles.

### 5.2. Main Results

**Objective Evaluation Results.** As shown in Table 2, the objective evaluation results demonstrate the superior performance of OmniAudio across multiple metrics on both YT360 and Sphere360 datasets: 1. **Non-Spatial Metrics**:

On the YT360 dataset, OmniAudio achieves FD of 92.57 and KL divergence of 1.64, outperforming the baselines

---

[5]For all comparisons, we use the frontal-view large FoV (120°) as the default perspective video. This choice aligns with real-world conditions where front-facing viewpoints capture main sound sources.

*Table 2.* Performance comparison between OmniAudio and the baselines on the Sphere360-Bench (in-distribution) and YT360 (out-of-distribution) test sets. We use objective metrics computing **FD**, **KL divergence**, $\Delta_{abs}\theta$, $\Delta_{abs}\phi$, and $\Delta_{Angular}$ between estimated DoA and ground truth, as well as subjective metrics including MOS for spatial audio quality (**MOS-SQ**) and video-audio alignment faithfulness (**MOS-AF**). We report the mean and standard deviation for MOS-SQ and MOS-AF. **+AS** denotes adding an audio spatialization component. For metrics with a downward arrow (↓), lower values represent better performance, while for metrics with an upward arrow (↑), higher values indicate better quality.

| Model | Params | FD↓ | KL↓ | $\Delta_{abs}\theta\downarrow$ | $\Delta_{abs}\phi\downarrow$ | $\Delta_{Angular}\downarrow$ | MOS-SQ↑ | MOS-AF↑ | Inf. Time |
|---|---|---|---|---|---|---|---|---|---|
| **In-distribution (Sphere360-Bench)** | | | | | | | | | |
| GT | - | - | - | - | - | - | 88.41±0.79 | 90.12±1.08 | - |
| Diff-Foley + AS | 0.94B | 331.05 | 3.56 | - | - | - | 69.87±0.84 | 71.12±1.36 | 2.40s |
| MMAudio + AS | 1.03B | 271.15 | 2.39 | - | - | - | 75.34±0.99 | 77.56±1.22 | 3.01s |
| ViSAGe (FoV) | 0.36B | 210.87 | 2.90 | 1.51 | 0.71 | 1.49 | 73.45±1.42 | 74.89±1.71 | 22.37s |
| ViSAGe (360) | 0.36B | 219.66 | 2.96 | 1.52 | 0.74 | 1.51 | 74.12±1.18 | 75.34±1.03 | 22.37s |
| **OmniAudio** | 1.22B | **88.30** | **1.58** | **1.36** | **0.52** | **1.28** | **84.67±1.06** | **87.23±0.98** | 0.92s |
| **Out-of-distribution (YT360-Test)** | | | | | | | | | |
| GT | - | - | - | - | - | - | 85.38±0.95 | 87.85±1.21 | - |
| Diff-Foley + AS | 0.94B | 361.65 | 2.22 | - | - | - | 67.21±0.95 | 70.34±1.76 | 2.40s |
| MMAudio + AS | 1.03B | 190.40 | 1.71 | - | - | - | 73.25±1.05 | 76.77±1.23 | 3.01s |
| ViSAGe (FoV) | 0.36B | 199.09 | 1.86 | 2.21 | 0.88 | 1.99 | 71.82±1.98 | 72.17±1.47 | 22.37s |
| ViSAGe (360) | 0.36B | 225.52 | 1.95 | 2.18 | 0.86 | 1.98 | 72.45±1.64 | 72.96±1.39 | 22.37s |
| **OmniAudio** | 1.22B | **92.57** | **1.64** | **1.27** | **0.53** | **1.27** | **80.37±0.91** | **83.49±1.01** | 0.92s |

Diff-Foley + AS (FD: 361.65, KL: 2.22) and MMAudio + AS (FD: 190.40, KL: 1.51). This indicates a substantial improvement in perceptual audio quality. Similarly, on the Sphere360 dataset, OmniAudio achieves FD of 88.30 and KL divergence of 1.58, surpassing Diff-Foley + AS (FD: 331.05, KL: 3.56) and MMAudio + AS (FD: 271.15, KL: 2.39). 2. **Spatial Audio Metrics**: OmniAudio consistently outperforms all baselines in spatial audio metrics. On the YT360 dataset, OmniAudio achieves $\Delta_{Angular}$ of 1.27, compared to 1.99 for ViSAGe (FoV), 1.98 for ViSAGe (360), and higher values for the other baselines. On the Sphere360 dataset, OmniAudio yields $\Delta_{Angular}$ of 1.28, outperforming ViSAGe (FoV) at 1.49 and ViSAGe (360) at 1.51. These results underscore the advantages of directly generating FOA audio from 360-degree video in accurately capturing spatial information, as opposed to generating FOA from perspective video inputs.

**Subjective Evaluation Results.** Evaluating 360V2SA models is inherently challenging due to the subjective nature of perceptual quality assessment. We conducted human evaluations and report the findings in Table 2. OmniAudio demonstrated the highest perceptual quality, attaining MOS-SQ and MOS-AF scores of 84.67 and 87.23, respectively. These scores highlight a clear preference by evaluators for the synthesized outputs of our model compared to baseline models, in terms of spatial audio perception and video-to-audio alignment. However, it is essential to acknowledge the inherent subjectivity in evaluating spatial audio quality, which contributes to the relatively large standard deviations observed in the MOS scores.

### 5.3. Case Study

In our qualitative analysis, we compare spectrograms of audio generated by OmniAudio and those produced by baselines, as illustrated in Figure 3. We make the following observations: (1) As demonstrated in case 1, MMAudio struggles to generate spatial audio when the source object is out of view. In contrast, OmniAudio effectively generates FOA audio even when the object moves behind the camera, highlighting the importance of 360-degree video in accurately recreating the 3D sound environment required for FOA audio. (2) The spectrograms produced by OmniAudio consistently exhibit higher fidelity and alignment with video compared to those generated by baselines. In case 2, OmniAudio maintains a more consistent musical rhythm, as evidenced by the performance of the percussionist in the video. These enhancements are indicative of OmniAudio's ability to better replicate the true acoustic scene, leading to more authentic and spatially precise audio outputs.

### 5.4. Ablation Study

We conduct an ablation study on the Sphere360 dataset to ensure consistency and reliability. Additional analyses are presented in Appendix G.

**Effect of Self-supervised Pre-training.** We evaluate efficacy of the self-supervised coarse-to-fine pre-training by comparing four configurations: (1) full model with coarse-to-fine pre-training (**coarse-to-fine**), (2) pre-training with spatial audio only (**w/ fine**), (3) pre-training with non-spatial audio only (**w/ coarse**), and (4) without any self-supervised

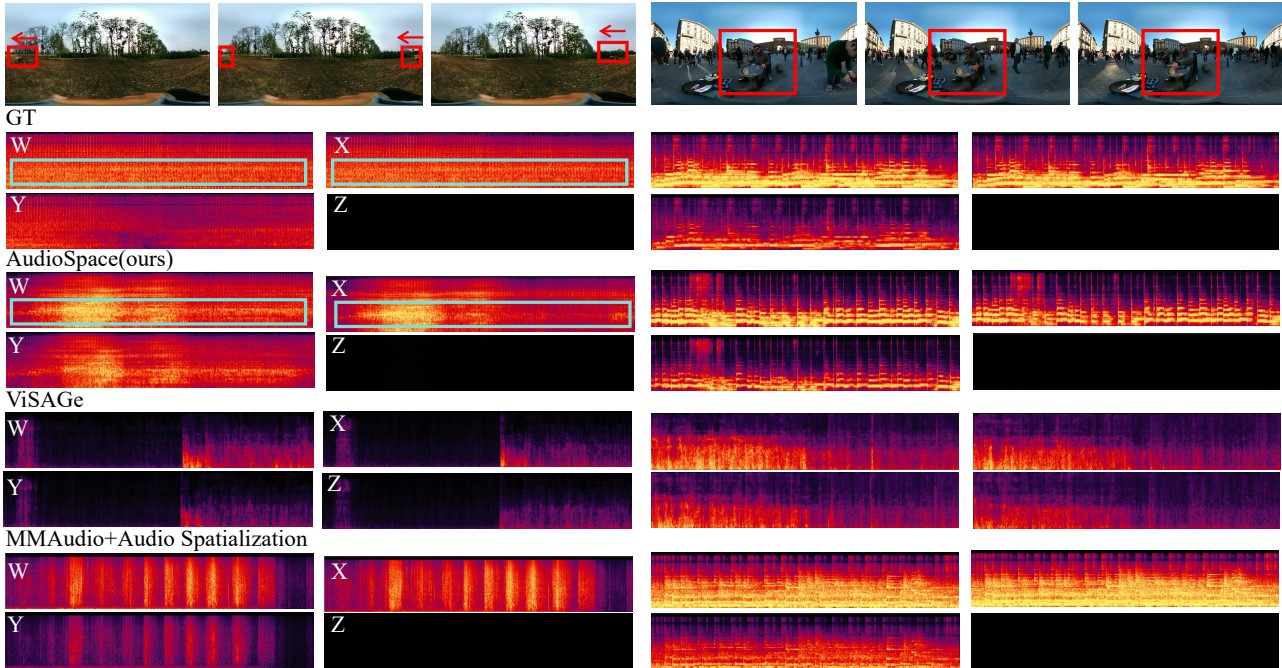

*Figure 3.* Qualitative Comparison. The first case on the left shows an agricultural machine moving behind, with the rectangular annotation indicating a decreasing trend in sound intensity in the GT audio. The second case on the right features a person playing a musical instrument. Since ViSAGe only generates 5-second audio, we concatenate the segments.

pre-training (**w/o PT**). As presented in Table 3, we can draw the following observation: (1) the coarse-to-fine approach consistently outperforms the other configurations across all metrics, highlighting the benefits of integrating both spatial and non-spatial audio datasets for pre-training. (2) the absence of pre-training and coarse stage results in notable degradation in performance, emphasizing the importance of self-supervised pre-training and non-spatial audio datasets in enhancing the generalization of the model.

*Table 3.* Effect of Self-supervised Pre-training.

| Model | FD↓ | KL↓ | $\Delta_{abs}\theta$↓ | $\Delta_{abs}\phi$↓ | $\Delta_{\textbf{Angular}}$↓ |
|---|---|---|---|---|---|
| coarse-to-fine | **88.30** | **1.58** | **1.36** | **0.52** | **1.28** |
| w/ fine | 97.57 | 1.82 | 1.36 | 0.57 | 1.28 |
| w/ coarse | 97.26 | 1.78 | 1.36 | 0.66 | 1.30 |
| w/o PT | 104.57 | 1.83 | 1.39 | 0.58 | 1.32 |

**Effect of Dual-branch Design.** We investigate the effectiveness of the dual-branch design by comparing it with models that utilize only the input perspective video (**w/ Per**), only the Equi-Angular Cubemap (a format of 360-degree video) (**w/ EAC**), and only equirectangular representations (**w/ ERP**). The results are presented in Table 4. We find that: (1) The dual-branch architecture outperforms single-branch models across all metrics, highlighting the benefit of integrating multiple representations for spatial audio generation. (2) While EAC outperforms ERP in single-branch settings, combining ERP and Per captures complementary spatial information more effectively, likely because EAC's division into six faces loses global context. (3) Panoramic videos significantly outperform perspective inputs in both spatial

and non-spatial metrics, demonstrating the advantages of full-view 360-degree video for generating immersive spatial audio. By combining perspective video with equirectangular representations in the dual-branch design, the model

*Table 4.* Effect of the Dual-branch Design.

| Model | FD↓ | KL↓ | $\Delta_{abs}\theta$↓ | $\Delta_{abs}\phi$↓ | $\Delta_{\textbf{Angular}}$↓ |
|---|---|---|---|---|---|
| ERP+Per | **88.30** | **1.58** | 1.36 | **0.52** | **1.28** |
| EAC+Per | 89.89 | 1.66 | **1.33** | 0.55 | 1.29 |
| w/ Per only | 88.80 | 1.87 | 1.41 | 0.59 | 1.33 |
| w/ EAC only | 93.37 | 1.84 | 1.37 | 0.57 | 1.30 |
| w/ ERP only | 97.83 | 1.87 | 1.35 | 0.59 | 1.28 |

**Impact of Model Size.** To evaluate the impact of the model size on OmniAudio's performance, we compare three model configurations: **Large (1.2B)**, **Medium (472M)**, and **Small (291M)**, using the Sphere360 dataset. We detail configurations of the different-sized models in Appendix D and use the Large model by default. We make the following observations from Table 5: (1) The Large model achieves the best performance across all metrics, including achieving the lowest FD and KL divergence. The results show that the capacity of a larger model enhances the generative quality and improves alignment with the ground truth distribution. (2) As the model size decreases from Large to Small, the performance degrades substantially. The Medium model yields moderately higher FD and KL results, while the Small model produces the highest divergence and spatial errors. These results highlight the difficulty in maintaining audio fidelity and spatial precision with smaller models, emphasizing the necessity of adequate model capacity for effective

audio-spatial generation.

*Table 5.* Impact of Model Size.

| Model Size | FD↓ | KL↓ | $\Delta_{abs}\theta\downarrow$ | $\Delta_{abs}\phi\downarrow$ | $\Delta_{\textbf{Angular}}\downarrow$ |
|---|---|---|---|---|---|
| Large | **88.30** | **1.58** | **1.36** | **0.52** | **1.26** |
| Medium | 104.19 | 1.82 | 1.36 | 0.60 | 1.28 |
| Small | 108.50 | 1.91 | 1.37 | 0.67 | 1.29 |

# 6. Conclusion

In this work, we introduced the novel task of generating spatial audio from 360-degree video, termed **360V2SA**. We demonstrated that traditional video-to-audio generation techniques often fell short in delivering immersive auditory experiences due to their reliance on non-spatial audio formats, which fail to capture the crucial spatial cues inherent in 360-degree video. By leveraging FOA audio, we aimed to capture the spatial cues necessary for accurately representing sound sources in three-dimensional environments. To facilitate research in this area, we created **Sphere360**, the first large-scale dataset specifically curated for 360V2SA. Our semi-automated pipeline for data collection and cleaning ensured high-quality paired video-audio data, which was essential for training robust models. We proposed the **OmniAudio** framework, which utilized self-supervised pre-training and a dual-branch architecture to effectively capture both local and global information from panoramic video inputs. Our experimental results demonstrated that OmniAudio achieved state-of-the-art performance across both objective and subjective metrics on the Sphere360 dataset. We envisage that our work could serve as a basis for future 360-degree video-to-spatial audio generation studies.

# Impact Statement

This paper advances the field of Machine Learning by presenting novel work in video-to-audio generation. This technology holds significant societal implications, both beneficial and detrimental, which are outlined below.

## 6.1. Positive Impacts

**Accessibility:** By generating audio from silent videos, this technology can make visual content more accessible to people with hearing impairments, thereby improving inclusivity in media consumption.

**Content Creation:** Creators can effortlessly add sound effects, background music, or narration to videos without requiring extensive audio editing skills, opening up new creative possibilities.

**Education and Training:** Educational videos can be enriched with automatically generated audio explanations, making learning experiences more engaging and accessible.

**Virtual Reality and Augmented Reality:** Realistic audio generated from visual input can significantly enhance immersion in virtual and augmented reality experiences.

**Historical Preservation:** Silent historical footage can be brought to life by adding generated audio based on visual cues, preserving valuable historical moments.

## 6.2. Negative Impacts

**Misinformation and Deepfakes:** The ability to generate realistic audio from video could be exploited to create fake news or manipulate audio content to misrepresent someone's words or actions, raising ethical concerns.

**Privacy Concerns:** There are risks of privacy violations if video-to-audio technology is used to generate audio from personal videos without consent.

**Job Displacement:** As AI-powered audio generation becomes more sophisticated, professionals in audio editing and sound design may face job losses.

**Copyright Issues:** Generating audio that closely resembles existing copyrighted music or sound effects could lead to legal complications.

## 6.3. Key Considerations

**Regulation and Transparency:** Developing guidelines and regulations to address the potential misuse of video-to-audio generation technology is essential.

**Ethical Development:** Researchers and developers should prioritize responsible development practices to mitigate potential harms and ensure

# Acknowledgements

The research was supported by NSFC (No.62206234) from Mainland China, Early Career Scheme (ECS-HKUST22201322) from Hong Kong RGC, and National Natural Science Foundation of China under Grant No.62222211 and No.U24A20326.

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

## Table of Contents

## A. Disclaimer on Copyright and Data Usage

The video data utilized in this study were sourced from the YouTube platform. All content is copyrighted by their respective creators and owners. The videos included in this research adhere to YouTube's terms of service and, where applicable, to Creative Commons licenses. Specifically, videos under the Creative Commons license have been appropriately attributed to the original authors in accordance with the license terms (CC BY 4.0).

For videos not governed by a Creative Commons license, we acknowledge that they are protected by copyright and are used exclusively for academic research purposes. No commercial use of these videos or content is intended. The use of these videos falls under the fair use doctrine for educational and research purposes, as permitted by copyright law.

# B. Dataset Construction Pipeline

## B.1. Data Crawling

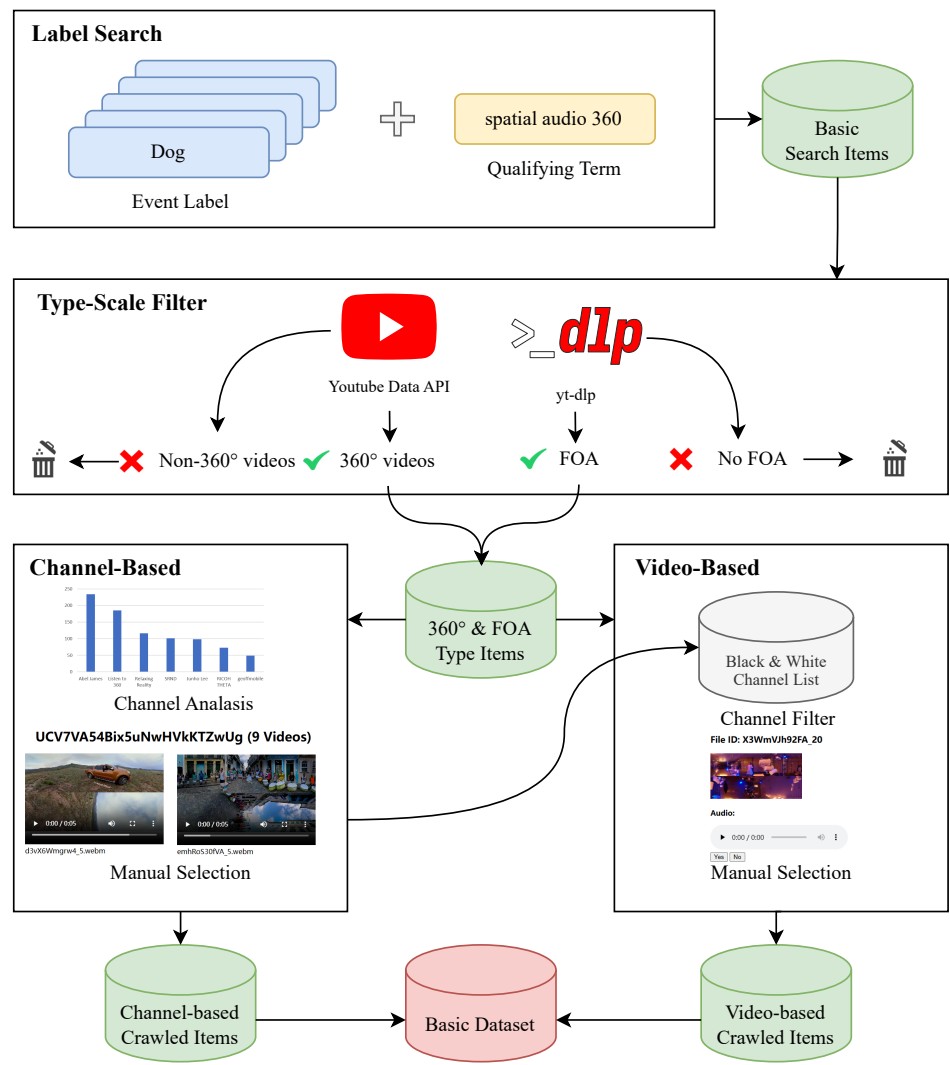

*Figure 4.* The process of dataset crawling

**Label Searching**   To identify relevant videos, we employ a keyword-based search strategy that targets entities commonly associated with 360° and FOA content. Our search keywords combine specific event labels to ensure class diversity with qualifying terms aimed at retrieving more 360° and FOA content. We derive our event labels from the ontology used in AudioSet (Gemmeke et al., 2017). To ensure the labels are simple, comprehensive, and likely to appear in video titles, we limit our selection to the first three levels of the hierarchical ontology. Through manual curation, we have compiled a list of 316 event labels. Throughout the process, we periodically re-evaluate the search results to refine the keywords, removing event labels that yield few or low-quality results to improve class diversity, and testing multiple qualifying terms to increase the proportion of 360° content. As a result, we selected "spatial audio 360" as the primary qualifier. This process led to the compilation of 316 search keywords in the format "[event label] spatial audio 360".

**Type-Scale Filter**   We filter out videos that lack either 360-degree visual content or spatial audio using the YouTube Data API (by checking the "projection" field within "contentDetails" to ensure it is set to "360") and yt-dlp (using the format filter "audio_channels" to exclude videos that do not support 4-channel audio for any player client). This approach enables us to identify videos that contain both 360° video and FOA audio.

**Channel-Based Crawling**    In the initial stage of crawling, we use a large-scale approach to filter relevant channels. We start by searching up to 5 pages (a maximum of 250 results) for each keyword and then filter out 360° and FOA entities to create a smaller search set. Next, we identify channels that appear more than three times within the 5-page search set. For each prominent channel, we randomly select 10 videos and manually filter for relevance. This process results in 124 relevant channels out of an initial 246. By collecting all videos from these channels, we compile a list of 1488 highly relevant videos. Additionally, we create blacklists (for totally irrelevant channels) and whitelists (for highly relevant channels) based on manual filtering in the first stage. These lists will be used in the next crawling stage to streamline the filtering process.

**Video-Based Crawling**    In the second stage, we expand the search scale to create a larger search set. Since entities beyond the first 5 search pages tend to be less relevant and contain fewer 360° and FOA items, we manually filter out irrelevant videos. By searching up to 12 pages for each keyword and excluding videos from the blacklists or whitelists established in the previous stage, we obtain 652 360° and FOA items, of which 438 are manually identified as relevant. Additionally, we experiment with other qualifying terms for searching, and the top two are detailed in Table 6. The overall details of the two-stage crawling are shown in Table 7, and the whole procedures are illustrated in Figure 4.

Finally, by adding 83K clips (total duration of 230.6 hours) successfully downloaded from 5128 videos in the YT-360 dataset, we obtained a total of 166.5K clips (approximately 462.5 hours) from 7179 videos.

*Table 6.* Details of Video-Based Crawling. Different qualifying terms are applied independently. Selected Videos refers to the number of videos included in the dataset.

| Qualifying Term | 360° Videos | After Channel Filter | 360° & FOA Videos | Videos Selected |
|---|---|---|---|---|
| spatial audio 360 | 5996 | 2036 | 652 | 438 |
| 3D audio 360 | 4353 | 2899 | 384 | 245 |
| **Total** | 8746 | 4356 | 848 | 563 |

*Table 7.* Outcome of the Two Stages. Videos Covered denotes the number of 360° videos in the search set (for Channel-Based Crawling, this refers to the total number of 360° videos from all checked channels). Videos Selected, Clips, and Total Duration indicate information of relevant videos manually identified as relevant.

| Stage | Videos Covered | Videos Selected | Clips | Total Duration |
|---|---|---|---|---|
| Channel-Based Crawling | 13733 | 1488 | 58.9K | 163.6h |
| Video-Based Crawling | 8746 | 563 | 24.6K | 68.3h |
| **Total** | 22479 | 2051 | 83.5K | 231.9h |

### B.2. Data Cleaning

**Stationary Videos**    To clean stationary videos, we measure the similarity between two frames at fixed intervals using the mean squared error (MSE). A certain threshold of similarity is set to classify frames as stationary. If a video segment contains more than 85% of stationary frames, it is classified as a stationary video and subsequently removed from the dataset.

**Silent Audio**    For cleaning silent audio, we divide the audio into several segments and calculate the maximum dBFS value across all channels for each segment. If this value falls below -35, we consider the segment to be silent. If the number of silent segments in the entire audio exceeds 90%, the audio is classified as silent and removed from the dataset.

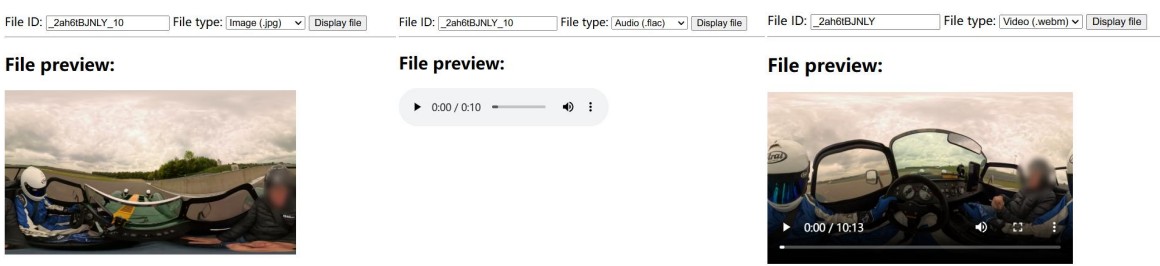

*Figure 5.* Manual inspection webpage.

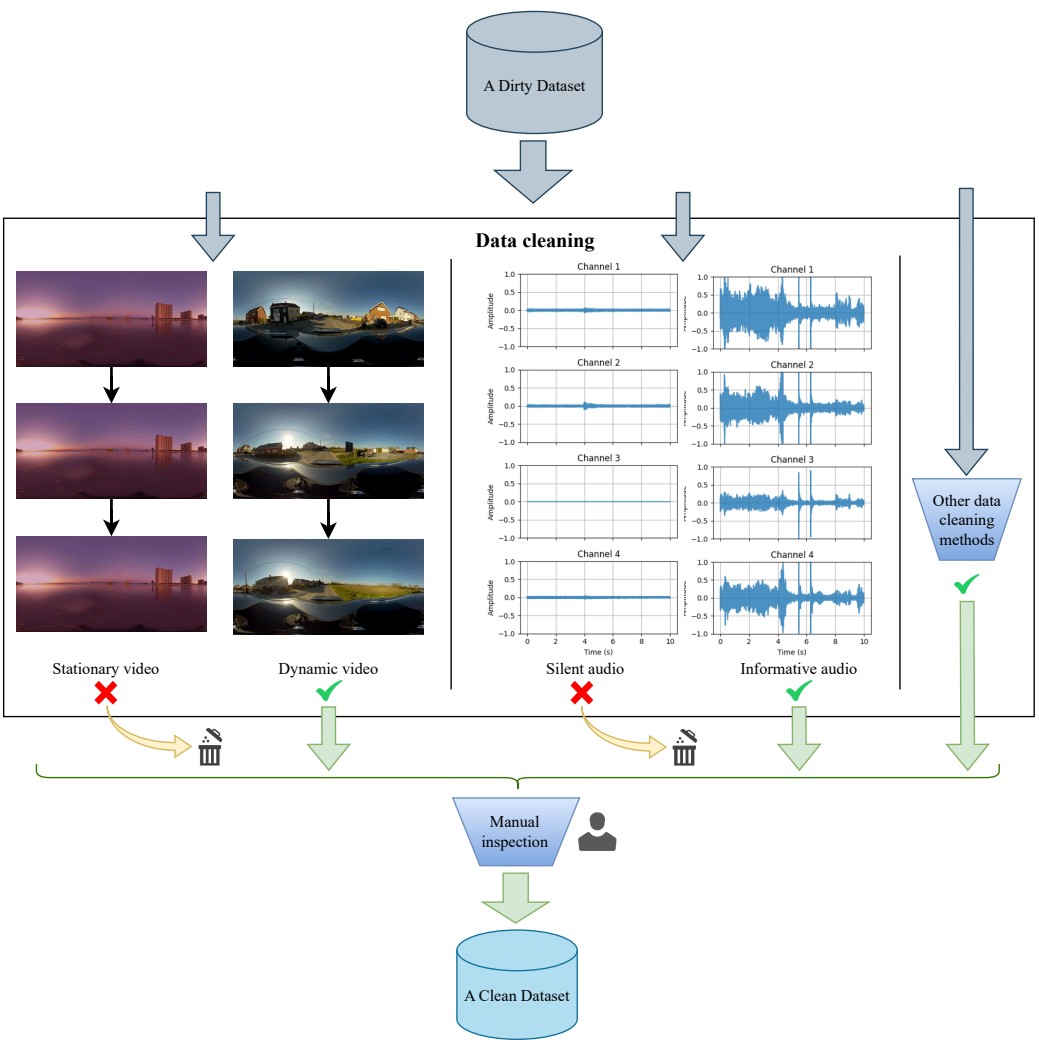

*Figure 6.* The process of dataset cleaning.

**Excessive Speech** For videos with excessive speech, we use SenseVoice (An et al., 2024) to detect the presence of speech and remove videos where the number of detected words exceeds 5. We do not remove all videos containing speech, as this approach allows us to retain videos that feature moderate levels of speech, which may still be valuable for the dataset. This ensures that videos with important context or minimal speech are not erroneously discarded, preserving a more balanced range of content.

**Audio-visual Mismatch** For cleaning based on audio-visual alignment, we use Imagebind (Girdhar et al., 2023) to assess the degree of alignment between video and audio. We stipulate that videos with a degree higher than 2 are considered qualified. Videos that do not meet the threshold are removed from the dataset, ensuring that only videos with a strong and coherent connection between audio and visual content are retained. This approach helps to eliminate videos where the audio-visual mismatch may hinder the quality of data analysis or model training.

**Manual Inspection** We develop a clear and user-friendly webpage (Figure 5) to manually inspect the effectiveness of the data cleaning process. We sample and verify the data that has been removed to ensure it truly does not meet the criteria, preventing valid data from being mistakenly deleted. Additionally, we sample the remaining data after cleaning to confirm it meets the required conditions. If any discrepancies are found during this process, we review the appropriateness of the cleaning thresholds and the accuracy of the cleaning process, restart the cleaning, and make manual adjustments when needed, to ensure the effectiveness of the cleaning process. The overall data cleaning process diagram is shown in Figure 6.

### B.3. Pipeline Statistics

During the cleaning process, following the specific methods mentioned above, we remove approximately 7,500 silent clips, 12,000 static clips, 34,000 clips with excessive human voices, and 23,500 clips with audio-visual mismatches (some of the clips removed during the cleaning process may overlap). The number of clips in each category of the cleaned dataset is shown in Figure 7.

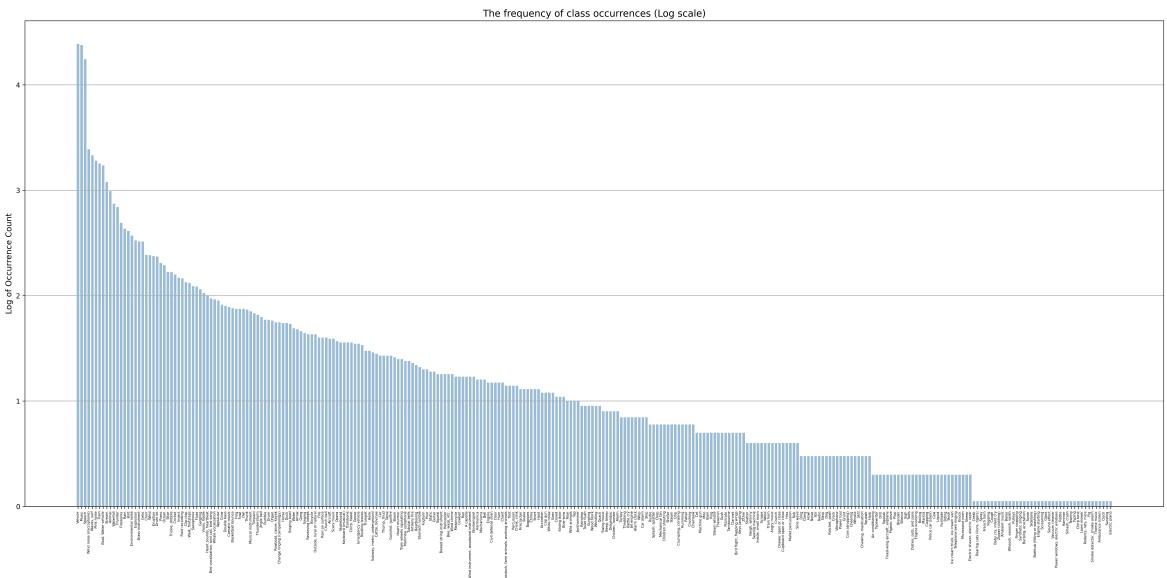

*Figure 7.* The bar chart shows the occurrence count of different classes in the cleaned dataset (displayed on a log scale).

## C. Dataset Comparisons and Benchmark Construction

**Dataset Related Works and Comparisons**   Several datasets (Morgado et al., 2020; Liu et al., 2023b) have been developed to support research in 360-degree video and spatial audio processing. REC-STREET (Morgado et al., 2018), YT-ALL (Morgado et al., 2018), and YT-360 (Morgado et al., 2020) provide large-scale 360-degree video collections with First-order Ambisonics (FOA) audio (with total durations of 3.5, 113, and 246 hours, respectively), although their primary focus is on audio-visual correspondence learning. STARRS23 (Shimada et al., 2023), while supporting spatial audio generation from 360-degree videos, contains only 200 clips (7.5 hours), limiting its applicability for deep learning approaches. More recently, YT-Ambigen (Anonymous, 2024) introduced a dataset of 102K clips (totaling 142 hours), specifically designed for spatial audio generation; however, it is limited to fixed field-of-view (FoV) videos rather than full 360-degree content. Our

*Table 8.* Comparison of Sphere360 with existing datasets. FoV and 360° respectively represent field-of-view videos and panoramic videos. NS, Bin, and FOA refer to non-spatial audio, binaural audio, and first-order ambisonics, respectively. "/" indicates that it is not explicitly mentioned in the paper.

| Dataset | #Clips | Clip duration | Video length | Video & Audio Type | Audio Generation | Contains a Benchmark | #Class |
|---|---|---|---|---|---|---|---|
| VGGSound (Chen et al., 2020) | 200K | 10s | 550h | FoV & NS | Yes | Yes | 300 |
| FairPlay (Gao & Grauman, 2019a) | 1.8K | 10s | 5.2h | FoV & Bin | No | / | / |
| OAP (Vasudevan et al., 2020) | 64K | 2s | 15h | 360° & Bin | No | / | ≥3 |
| REC-STREET (Morgado et al., 2018) | 123K | 0.1s | 3.5h | 360° & FOA | No | Yes | / |
| YT-ALL (Morgado et al., 2018) | 3976K | 0.1s | 113h | 360° & FOA | No | Yes | ≥8 |
| YT-Ambigen (Anonymous, 2024) | 102K | 5s | 142h | FoV & FOA | Yes | / | >300 |
| STARRS23 (Shimada et al., 2023) | 0.2K | <10min | 7.5h | 360° & FOA | Yes | Yes | 13 |
| YT-360 (Morgado et al., 2020) | 89K | 10s | 246h | 360° & FOA | No | / | 32 |
| **Sphere360** | **103K** | **10s** | **288h** | **360° & FOA** | Yes | Yes | 288 |

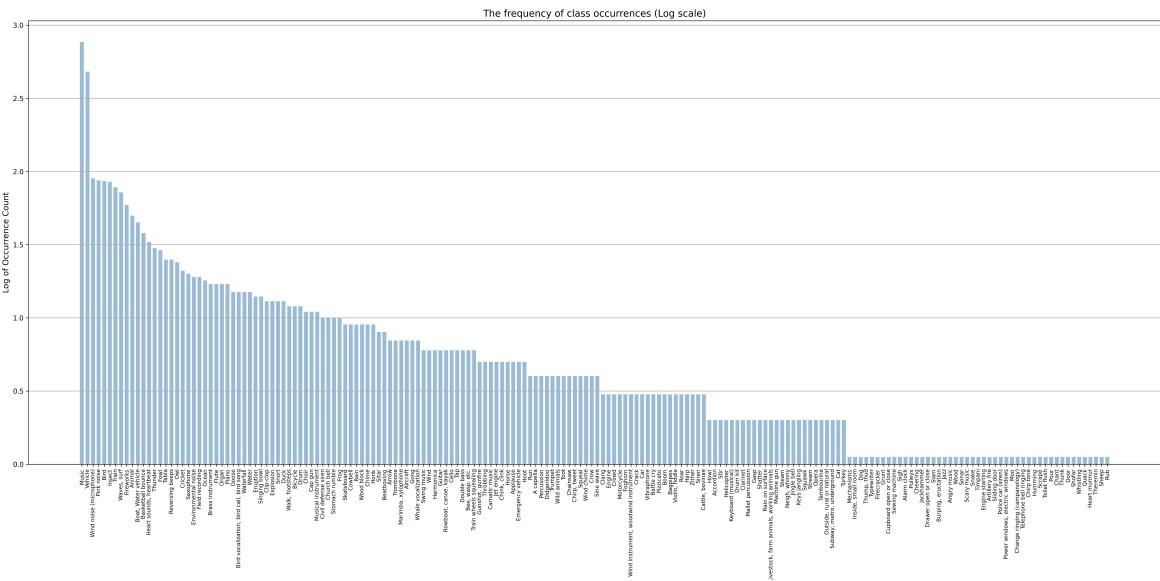

*Figure 8.* The bar chart shows the occurrence count of different classes in the benchmark dataset (displayed on a log scale).

Sphere360 dataset offers over 103K high-quality 360-degree video clips with FOA audio, totaling 288 hours in duration, specifically curated for spatial audio generation through meticulous data collection and cleaning pipelines. A comparison of Sphere360 with existing datasets is shown in Table 8.

**Benchmark**    Currently, there is a lack of an accessible benchmark for 360-degree video-to-spatial audio generation. To address this gap, we construct a benchmark encompassing about 180 distinct audio events using our semi-automated data-cleaning process. During the construction of the benchmark, we first label and categorize the dataset based on the audio. Then, we apply the semi-automated cleaning process for an initial screening with lower standards to remove low-value video segments. Next, we remove the video classes that are not of interest from the remaining dataset and select several video segments with the highest confidence for each class. During this selection, we try to retain complete video clips to better assess the generalization capabilities of the models. Following this, we utilize our designed web platform for manual verification, where samples failing to meet the standards are rejected and reprocessed. We iterate this process, adjusting the parameters used for filtering at each step, ultimately selecting around 3,000 high-quality clips and covering about 180 events. The class occurrence frequency of the benchmark dataset is shown in Figure  8.

## D. Model Configurations and Architecture

**Model Configurations**    For VAE training, we initialize our Spatial VAE using the VAE model weights trained on stereo data provided by Stability AI [6]. We employ mixed precision training with a batch size of 144 across 24 A800 GPUs for 500,000 steps. Subsequently, following Evans et al. (2024), we freeze the VAE encoder and train the VAE decoder with a latent mask ratio of 0.1 for an additional 300,000 steps. We use AdamW (Loshchilov & Hutter, 2019) as the optimizer, setting the generator learning rate to 3e-5 and the discriminator learning rate to 6e-5.

In the self-supervised pre-training phase, we apply a mask with a conditioning probability $p_{cond}$ of 0.1. We utilize exponential moving average and automatic mixed precision for 100,000 steps on 8 A100 GPUs, with an effective batch size of 256. For the Video-Guided fine-tuning stage, we similarly apply exponential moving average and automatic mixed precision for 50,000 steps on 8 A100 GPUs, maintaining an effective batch size of 256. AdamW remains our optimizer of choice, with a learning rate set at 5e-5.

**Variational Autoencoder**    The spatial VAE is initialized with pre-trained weights from a stereo VAE. To reconstruct FOA audio, we train the VAE using a weighted four-channel multi-resolution STFT loss. Additionally, we apply a KL divergence

---

[6]https://github.com/Stability-AI/stable-audio-tools

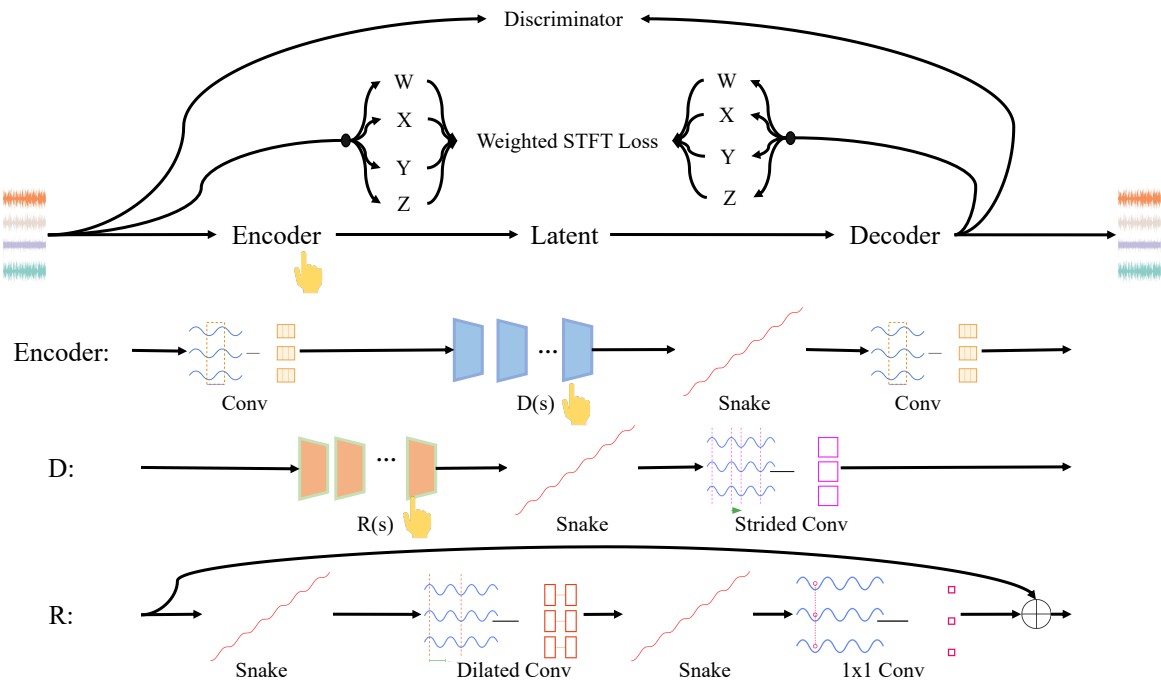

*Figure 9.* The architecture of autoencoder. D and R are designations for specific structures.

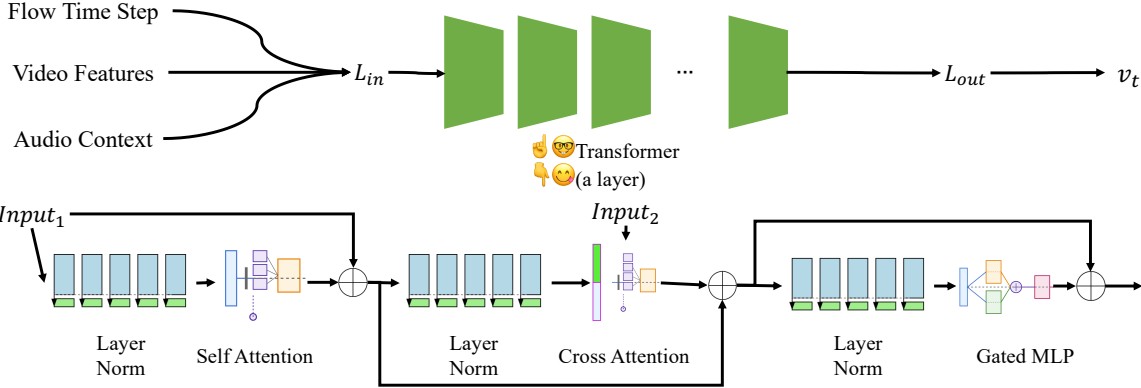

*Figure 10.* The architecture of diffusion transformer.

loss to the VAE bottleneck and a discrimination loss to enhance high-fidelity audio reproduction. The architecture of the VAE is illustrated in Figure 9.

**Diffusion Transformer**   The flow component employs a Diffusion Transformer (DiT) with an embedding dimension of 1536. It comprises 24 layers and 24 attention heads, with local and global conditioning dimensions of 768 and 1536, respectively. The transformer operates by projecting condition tokens and adheres to a continuous transformer architecture. The overall architecture is illustrated in Figure 10, while detailed configurations for different model scales are provided in Table 9. Given the scale of the existing dataset, we believe that a model with 1.2 billion parameters is already sufficiently large, and therefore, we did not further increase the capacity of our model.

*Table 9.* Diffusion Transformer Configurations at Different Model Size.

| Model Scale | Embedding Dimension | Depth | Attention Heads | Condition Token Dimension | Global Condition Dimension | Total Parameters |
|---|---|---|---|---|---|---|
| Large | 1536 | 24 | 24 | 768 | 1536 | 1.2B |
| Medium | 1024 | 16 | 16 | 512 | 1024 | 472M |
| Small | 768 | 12 | 12 | 384 | 768 | 291M |

# E. Evaluation

### E.1. Objective Metrics

The **Fréchet Distance (FD)** is used to measure the similarity between the feature distributions of generated and reference audio. A lower FD indicates that the generated audio is closer to the reference in terms of feature distribution ((Kilgour et al., 2018), (Copet et al., 2024)). We use the OpenL3 feature space for audio projection ((Cramer et al., 2019), (Evans et al., 2024)).

The **Kullback-Leibler (KL) Divergence** measures the difference between the label distributions of generated and reference audio. A lower KL indicates better semantic alignment between the generated and reference audio. The $\text{KL}_{\text{passt}}$ metric utilizes the PaSST model, an audio tagger trained on the AudioSet dataset ((Koutini et al., 2021)), to compute the KL divergence ((Copet et al., 2024)).

To assess the spatial accuracy of FOA (First-Order Ambisonic) audio, we calculate the intensities $I_x = \text{mean}(W \cdot X)$, $I_y = \text{mean}(W \cdot Y)$, and $I_z = \text{mean}(W \cdot Z)$ for the directional channels and report three key metrics ((Heydari et al., 2024)):

**Theta Error** ($\Delta_{abs}\theta$) measures the difference between the ground truth azimuth ($\theta$) and the estimated azimuth ($\hat{\theta}$). The azimuth is the angle in the horizontal plane, defined as:

$$\theta = \tan^{-1}\left(\frac{I_y}{I_x}\right)$$

The azimuth error is computed using the circular difference method ((Heydari et al., 2024)):

$$\Delta_{abs}\theta = \min\left(|\theta - \hat{\theta}|, 2\pi - |\theta - \hat{\theta}|\right)$$

**Phi Error** ($\Delta_{abs}\phi$) quantifies the difference between the ground truth elevation ($\phi$) and the estimated elevation ($\hat{\phi}$). The elevation angle is calculated as:

$$\phi = \tan^{-1}\left(\frac{I_z}{\sqrt{I_x^2 + I_y^2}}\right)$$

The elevation error is computed as the absolute difference between the ground truth and the estimated value:

$$\Delta_{abs}\phi = |\phi - \hat{\phi}|$$

**Spatial-Angle Error ($\Delta_{abs}$Spatial-Angle)** quantifies the difference between the ground truth and estimated directions of arrival (DoA). The spatial angle $\Delta$Spatial-Angle is calculated using the following equations:

$$a = \sin^2\left(\frac{\Delta\phi}{2}\right) + \cos(\phi) \cdot \cos(\hat{\phi}) \cdot \sin^2\left(\frac{\Delta\theta}{2}\right)$$

$$\Delta_{abs}\text{Spatial-Angle} = 2 \cdot |\arctan 2\left(\sqrt{a}, \sqrt{1-a}\right)|$$

Where $\Delta\theta$ and $\Delta\phi$ represent the linear and circular differences for azimuth and elevation, and $\phi$ and $\hat{\phi}$ represent the ground truth and estimated elevations.

The lower the values of these three metrics, the better the generation quality.

### E.2. Subjective Metrics

To probe spatial audio quality, we conduct the MOS (mean opinion score) tests and explicitly instruct the raters to "focus on examining the audio quality, naturalness, spatiality, and overall preference.". The testers present and rate the samples, and each tester is asked to evaluate the subjective naturalness on a 20-100 Likert scale.

To probe video-audio alignment, human raters are shown a spatial audio and a 360-degree video and asked "Does the spatial audio align with 360-degree video faithfully?". They must respond with "completely", "mostly", or "somewhat" on a 20-100 Likert scale.

Because of the characteristics of 360-degree videos and spatial audio, we recruit 15 participants in person rather than crowdsourcing like Amazon Mechanical Turk. We randomly select 50 samples from our Sphere360-Bench for each annotator.

### E.3. Spatial Audio Format

Spatial audio is a technology that captures and reproduces sound in such a way that it mimics the natural experience of hearing, providing listeners with a full 360-degree auditory environment. This involves accurately encoding sound direction, distance, and amplitude to create immersive and realistic audio experiences. In this context, FOA is a popular format due to its ability to effectively balance spatial resolution with computational simplicity. FOA achieves this using four specific channels labeled W, X, Y, and Z. Each channel serves a unique function: the W channel captures overall sound pressure from all directions, the X channel differentiates sounds from the front and back, the Y channel distinguishes left from right, and the Z channel encodes vertical audio information, such as sounds coming from above or below.

### E.4. Audio Spatialization Component

Given the ground-truth direction $(\phi, \theta)$ calculated from the previous section E.1, we can spatialize a mono audio signal into First-Order Ambisonics (FOA) format. Following (Zotter & Frank, 2019), the encoding process converts a mono sound source $s(t)$ into FOA channels as: $W = \frac{1}{\sqrt{2}}s(t)$, $X = \cos\theta\cos\phi s(t)$, $Y = \sin\theta\cos\phi s(t)$, and $Z = \sin\phi s(t)$. To spatialize our generated mono audio, we directly apply this encoding with $s(t)$ being our input signal, thereby positioning the sound in the corresponding direction in the ambisonic domain.

## F. Limitations and Future Work

**Limitations**    While our dataset demonstrates strong performance for the majority of test cases, it is worth noting that some samples, derived from real-world scenarios, contain an unusually high number of sound-emitting objects, as illustrated in Figure 11. In such cases, OmniAudio struggles to accurately identify event types, leading to errors such as misclassifying musical instrument sounds as applause.

**Future Work**    We plan to explore techniques for better understanding 360-degree videos with multiple targets. Additionally, our current cleaned dataset comprises 100,000 samples, which remains insufficient for robust real-world 360V2SA. To address this, we will leverage our semi-automated pipeline to continuously collect and expand the dataset, thereby advancing progress in this field.

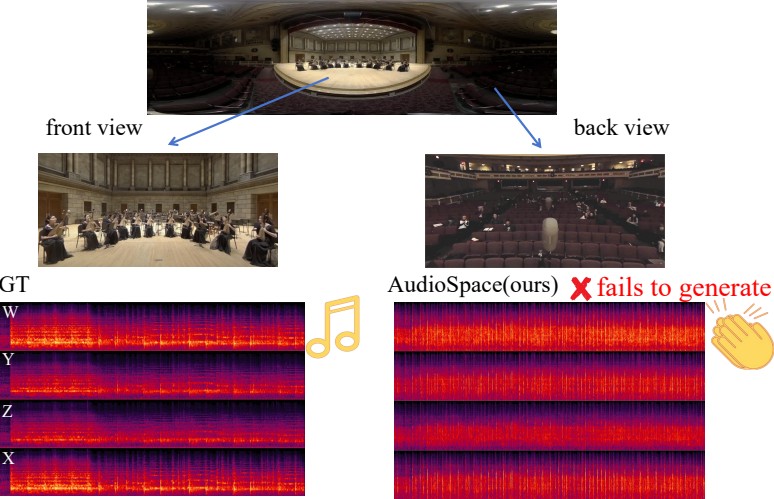

*Figure 11.* An failure case of OmniAudio. In this case, the front view shows an instrumental ensemble and the back view shows the audience. The sound source in the GT is the instrument sound from the front, but the model mistakenly identifies the sound source as applause from the audience behind.

## G. Additional Quantitative Results

**Impact of different Classifier-Free Guidance Scale (CFG-Scale)**    Testing with different CFG-Scale settings, the results are shown in Figure 12, illustrating how FD and $\Delta_{abs}$ Spatial-Angle change.

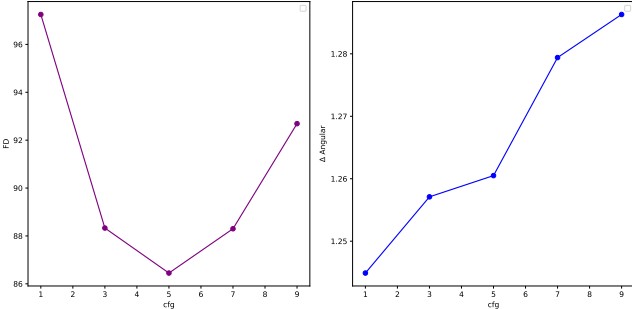

*Figure 12.* Variation of FD and $\Delta_{abs}$ Spatial-Angle under Different CFG-Scale Settings

Balancing the performance of FD and spatial results, we choose CFG-Scale = 5.

**Effect of Spatial VAE**    The results of the Variational Autoencoder (VAE) model in terms of STFT, MEL distances, FD, and KL divergence are presented in Table 10. A comparison is made between the proposed model (ours), descript-audio-codec (Kumar et al., 2024), and a non-spatial audio VAE model. Better performance is indicated by lower values of STFT, MEL distances, FD, and KL. Considering all these metrics, our model demonstrates superior overall performance, outpacing the others.

*Table 10.* Performance of VAE model in terms of STFT and MEL distances. For metrics with a downward arrow (↓), lower values represent better performance.

| Model | STFT Distance↓ | MEL Distance↓ | FD↓ | KL↓ |
|---|---|---|---|---|
| Ours | 0.82 | **1.48** | **94.15** | **0.31** |
| Descript-Audio-Codec | **0.71** | 4.74 | 123.41 | 0.62 |
| Non-Spatial Audio VAE | 4.12 | 6.65 | 272.89 | 2.12 |

**More Analysis of Dual-Branch Strategies**   To further investigate the effectiveness of our dual-branch approach, we conducted additional experiments comparing various FOV-cut strategies. Our original design employs a 120° frontal view to focus on prominent sound sources while incorporating panoramic features as contextual conditioning. We evaluated three alternative FOV-cut strategies to replace the local FOV video:

- Hexadirectional 360° cuts (ERP+6cuts): Capturing views from front, back, left, right, up, and down directions.

- Quadrant cuts (ERP+4cuts): Capturing views from front, left, right, and back directions.

- Bipolar cuts (ERP+2cuts): Capturing views from front and back directions.

Table 11 presents the results of this comparison.

*Table 11.* Comparison of Dual-Branch Methods

| Dual-Branch Method | FD ↓ | KL ↓ | $\Delta_{abs}\theta$ ↓ | $\Delta_{abs}\phi$ ↓ | $\Delta_{Angular}$ ↓ |
|---|---|---|---|---|---|
| ERP+Front | **88.30** | **1.58** | **1.36** | 0.52 | 1.28 |
| ERP+2FOV Cuts | 95.84 | 1.77 | 1.37 | 0.55 | 1.30 |
| ERP+4FOV Cuts | 92.89 | 1.65 | 1.36 | **0.51** | 1.27 |
| ERP+6FOV Cuts | 90.16 | 1.59 | 1.37 | 0.52 | **1.26** |

The results demonstrate that our original dual-branch strategy (ERP+Front) still achieves the best audio quality, as reflected in the FD and KL metrics. Moreover, it achieves similar spatial metrics compared to the multiple FOV view approaches. This analysis supports our choice of using a single frontal view combined with equirectangular projection in our final model design.

## H. Additional Qualitative Results

We include more qualitative results in this section, as shown in Figure 13, 14, and  15.

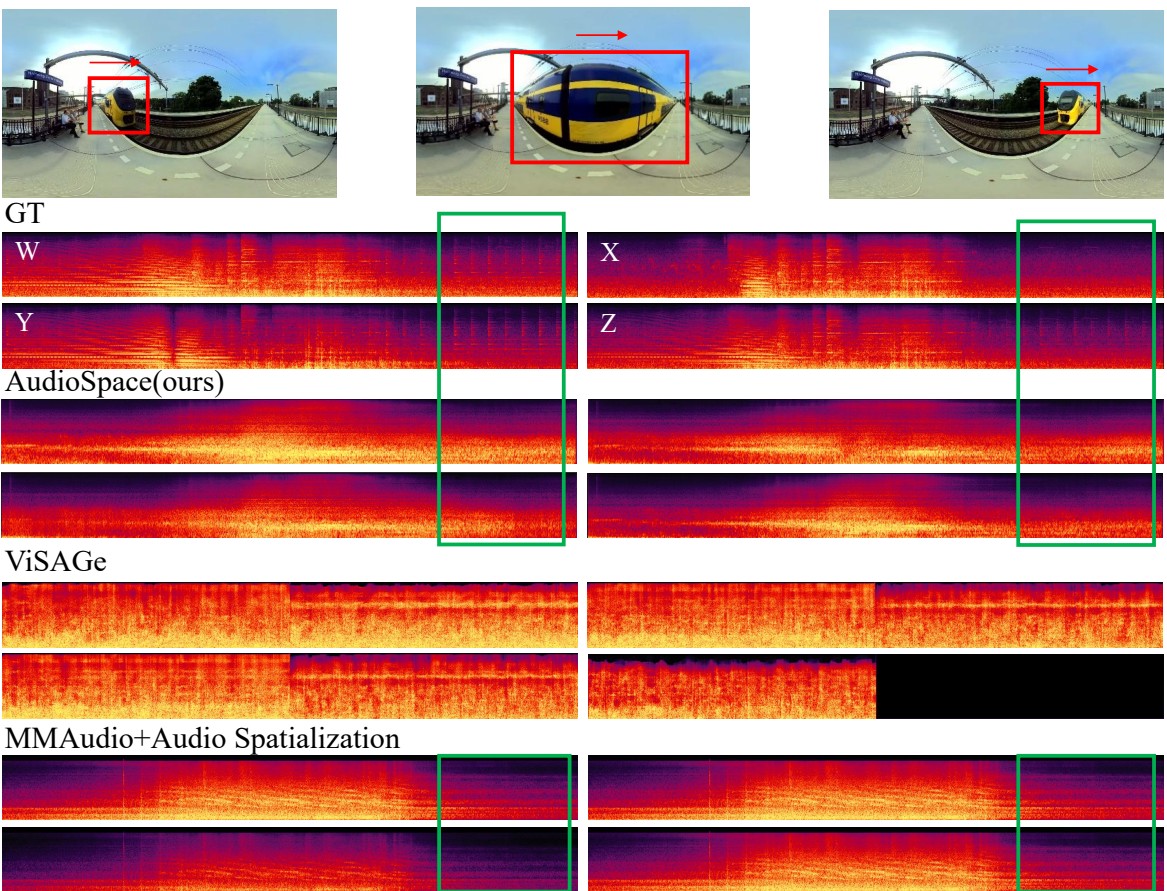

*Figure 13.* Additional Quantitative Results. This case shows a train passing by. The rectangular annotation indicates that the audio generated by our model continues to capture the sound of the train leaving the frontal perspective, even after it has passed, while the audio generated by other models almost entirely fades once the train moves out of the frontal view.

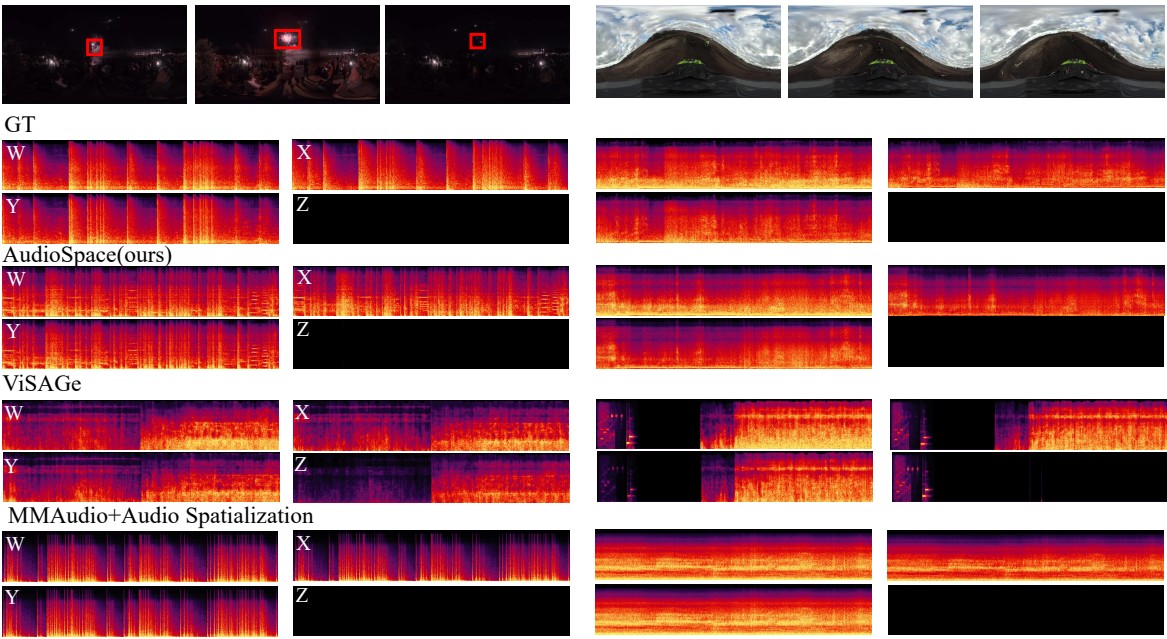

*Figure 14.* Additional Quantitative Results. The case on the left shows a continuous display of fireworks rising into the sky and exploding. The case on the right depicts several motorcycles chasing each other on a dirt road, with intense wind and engine sounds.

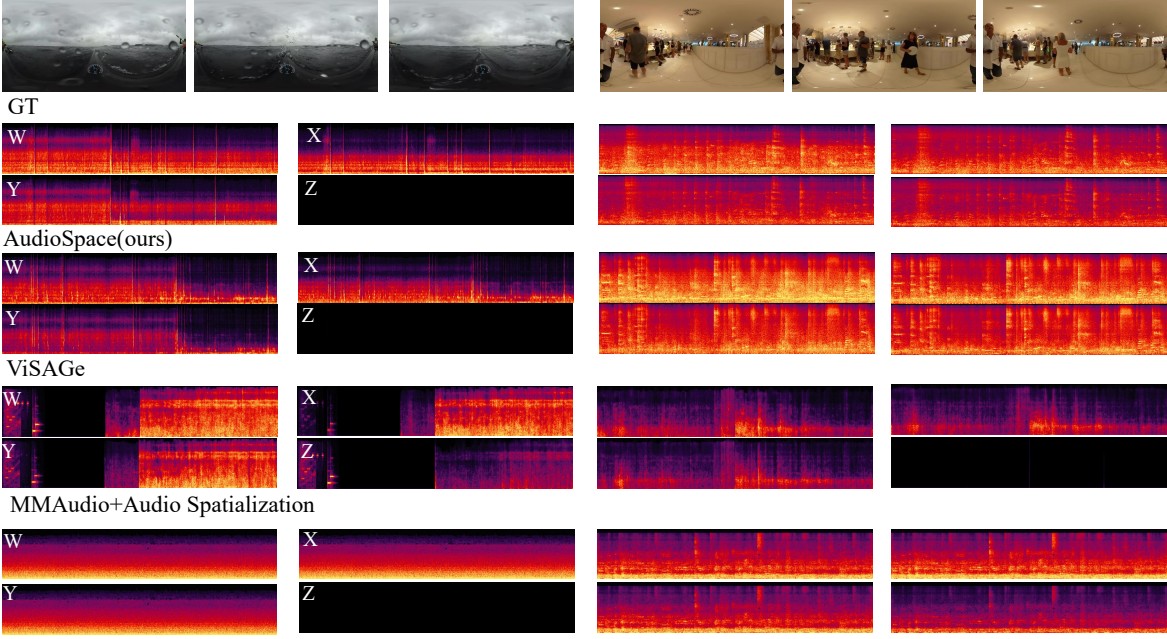

*Figure 15.* Additional Quantitative Results. The case on the left shows a camera mounted on a boat navigating through the waves, with the bow plunging into the water and splashing onto the screen. The case on the right shows the viewpoint moving through a noisy crowd in an indoor environment.

