# OpenReview forum: "OmniAudio: Generating Spatial Audio from 360-Degree Video"
_ICML.cc/2025/Conference — ICML 2025 poster_

### Official Review · Reviewer_cQFE · 2025-02-24

**Overall Recommendation:** 3

**Summary:**

This work created Sphere360, a real-world dataset for realistic 3D audio reproduction. An efficient sem-automated pipeline for collecting and cleaning paired video-audio data is established. The challenges of the created task are clearly described. The demos are interesting. Code and datasets will be made publicly available.

**Claims And Evidence:**

Yes. The dual-branch framework which utilizes panoramic and FoV video inputs is verified on the created Sphere360 dataset. The claimed contributions are well supported.

**Essential References Not Discussed:**

No. The related work analysis is comprehensive.

**Experimental Designs Or Analyses:**

Yes. The experimental results and comprehensive analyses provide evidences of the effectiveness of the proposed solution.

**Methods And Evaluation Criteria:**

Yes. The created Sphere-Bench provides a credible benchmark.

**Other Comments Or Suggestions:**

If it is possible, please consider conducting a user study to compare the quality of the generated spatial audio data.

The term of "FOV video inputs" could be revised to "perspective video". FOV means the field of view, which could also be associated to panoramic data with a large FOV.

**Other Strengths And Weaknesses:**

1. If it is possible, please consider conducting some experiments with different FoV setups for comparison to help understand the most suitable configuration of the framework depicted in Fig. 2.
2. The computational complexity results like FLOPs/MACs, the number of parameters, and the training/inference time of the proposed method could be presented to help understand the efficiency of the proposed solution.
3. It would be nice to discuss some limitations of the presented work and point out some future work directions.
4. In the related work section, it would be nice to also discuss some research works on panoramic video processing such as panoramic semantic segmentation and panoramic generation, etc.

Most of the concerns have been addressed in the rebuttal. The reviewer would like to maintain the positive rating.

**Questions For Authors:**

Would you consider your proposed method with more recent flow-matching strategies to help better understand the effectiveness of your solution?

**Relation To Broader Scientific Literature:**

Yes. The method is relevant to autonous driving and augmented reality.

**Theoretical Claims:**

Yes. The usage of the equirectangular representation and the extraction of the FoV video follow standard projections.

---

> ### Author Rebuttal · Authors · 2025-04-01
>
> We sincerely appreciate your recognition of our demo page and our dataset and all your valuable feedback. We plan to open-source the codebase in April 2025 to facilitate community-driven improvements in this direction and welcome the reviewer's specific recommendations on cutting-edge techniques worthy of prioritized exploration. We hope our response fully addresses your concerns and questions.
>
> ## Open-Source Sphere360 Dataset
>
> Please check our **Response to Reviewer nf2N** under **Open-Source Sphere360 Dataset**.
>
> ## Updated Table 2 with Model Size, Inference Time, and TFLOPS
>
> We sincerely apologize for the typos in Table 2 in the submission. Please check our **Response to Reviewer Xvna** under **Claims that AudioSpace achieves SOTA performance** As shown in the updated Table 2, **AudioSpace indeed achieves state-of-the-art performance on both in-domain Sphere360 and out-domain YT360-Test.** We also added the number of parameters, inference time, and TFLOPS into comparison. Although AudioSpace has slightly more parameters than MMAudio+AS, **it achieves notably faster inference than all baselines.**
>
> ## Extended V2A Validation
> We further compared AudioSpace with baselines on the perspective video-to-stereo audio (V2A) generation task on VGGSound. Under the same multimodal setting as MMAudio (i.e., + Text modality), **AudioSpace+Text notably outperforms SOTA MMAudio and sets new SOTA performance on the traditional V2A task**, in addition to the new 360V2SA task.
>
> | Model|Params|FD|KL|
> |-------------------------|--------|--------|--------|
> | Diff-Foley              | 859M   | 324.88 | 3.03   |
> | Seeing-and-Hearing      | 415M   | 261.60 | 2.30   |
> | Frieren                | 159M   | 80.69  | 2.83   |
> | MMAudio                 | 1.0B   | 43.26  | 1.56   |
> | AudioSpace (ours)       | 1.2B   | 34.56  | 1.64   |
> | **+ Text modality** | 1.2B   | **33.24** | **1.40** |
>
> ## More Ablation Study on Dual-Branch Design
> Thank you for the insightful suggestion for more FOV settings. We attach more dual-branch experimental results in Response to Reviewer nf2N under [Combining Equirectangular Projection with Multiple FOV Cuts]. **They further demonstrate that our "Frontal + ERP" dual-branch design outperforms other dual-branch variants.** We will add these experimental results and analyses to the revised paper.
>
> ## Limitation and Future Work
>
> Thank you for the kind reminder. Please refer to Appendix F for discussions of limitations and future work.
>
> ## Related Works on Panoramic Video Processing
>
> We greatly appreciate your guidance in strengthening the contextualization of our work. In the revised paper, we will add discussions on advancements in panoramic video processing into Section 2 Related Work.
>
> ## User Study
>
> In this work, we employ human evaluation based on the Mean Opinion Score (MOS) to quantitatively assess both spatial audio quality (MOS-SQ) and video-audio alignment faithfulness (MOS-AF), as detailed in Section 5.1. Additionally, we present a comprehensive case study in Section 5.3 to comparatively analyze the generated outputs of AudioSpace against baseline methods. Supplementary qualitative results are included in Appendix H. We will further expand it with additional case studies in the revised paper. To enhance reproducibility and community accessibility, we are planning to deploy an online interactive demo platform via Gradio and Hugging Face Spaces for real-time generation and evaluation.
>
> ## Usage of Terminology
>
> Thank you for this suggestion. We agree that "perspective video" more accurately conveys the conventional narrow-FOV nature of the inputs compared to panoramic formats. In the revised paper, we will replace all instances of "FOV video" with "perspective video", and add a footnote in Section 3.1 clarifying that "Perspective video" denotes standard rectilinear projections with ≤120° horizontal FOV, distinct from 360° equirectangular formats.
>
> ## The Integration of more recent Flow-matching strategies
> We sincerely appreciate the reviewer's constructive suggestion regarding flow-matching strategies. The proposed coarse-to-fine self-supervised pre-training and dual-branch design are agnostic to flow matching strategies.
> Our current framework achieves state-of-the-art performance while maintaining the fastest inference speed in both 360V2A and V2A tasks. While our experiments demonstrate that the existing configuration optimally balances accuracy and efficiency for target scenarios, we fully agree that deeper integration of advanced flow matching strategies could unlock further theoretical insights.
> We plan to open-source the codebase in April 2025 to facilitate community-driven improvements in this direction, and welcome the reviewer's specific recommendations on cutting-edge techniques worthy of prioritized exploration.

---

> > ### Comment · Reviewer_cQFE · 2025-04-02
> >
> > The reviewer would like to thank the authors for their rebuttal and responses. Many of the concerns have been addressed. The usage of terminology and more ablations on the dual-branch design and validations should be updated in the final version. The reviewer would like to maintain the positive rating of weak accept.

---

> > > ### Author Response · Authors · 2025-04-04
> > >
> > > We sincerely appreciate your time, expertise, and constructive engagement throughout the review process. We are deeply grateful for your recognition of our efforts to address the concerns raised earlier. Your valuable feedback has been instrumental in significantly enhancing the quality of our work.
> > >
> > > Thank you very much for confirming that **"Many of the concerns have been addressed."**. We will update the usage of terminology and more ablations on the dual-branch design and validations, as well as other mentioned revisions, in the revised paper.
> > >
> > > To further facilitate open research, community engagement, and enhance the quality of our paper, we have completed the following works:
> > > 1. **Inference Code Release**
> > > We have open-sourced the inference code via the anonymous repository: https://anonymous.4open.science/r/Audiospace-1348/. Due to the model size (>10GB) and anonymity constraints, we are currently exploring feasible methods to share the pre-trained model weights. We will update the repository promptly once we resolve the issue of anonymously releasing large models. Additionally, we are actively organizing the training code and plan to release it by the end of April 2025.
> > > 2. **Enhanced Dataset Documentation**
> > > The Sphere360 dataset repository (https://anonymous.4open.science/r/Sphere360-CF51/) has been updated with enhanced documentation, including clearer usage guidelines and dataset structure descriptions.
> > > 3. **Manuscript Revisions**
> > > All suggestions and experimental results discussed during the rebuttal phase will be carefully incorporated into the revised manuscript to ensure a comprehensive presentation of our contributions.
> > >
> > > **We are more than happy to provide further responses to any additional questions or concerns you might have before the author response deadline**.
> > >
> > > We respectfully ask if you would consider increasing your original Overall Recommendation 3:Weak accept,  based on our responses and efforts to address all concerns and questions and enhance the paper, as we aim to fully address all feedback provided. Should any further clarifications or adjustments be needed, please feel free to share your concerns—we are fully committed to addressing them promptly.

---

### Official Review · Reviewer_nf2N · 2025-03-10

**Overall Recommendation:** 4

**Summary:**

This paper addresses a novel task called 360V2SA, which involves generating First-order Ambisonics (FOA) spatial audio from 360-degree videos. To tackle this challenge, the authors introduce a new dataset called Sphere360, containing more than 100k clips of real-world 360-degree videos paired with their FOA audio. They also propose AudioSpace, a dual-branch flow-matching framework that combines both panoramic and field-of-view (FOV) video representations. The model is first pre-trained in a self-supervised manner using both spatial (FOA) and non-spatial audio, and is then fine-tuned to generate high-quality spatial audio. Experimental results demonstrate that AudioSpace outperforms several baseline models.

## Update after rebuttal
After the discussion, I raised my rating since the authors addressed my concerns.

**Claims And Evidence:**

The claims made by the authors are largely supported by the experiments on Sphere360 and YT360. They demonstrate meaningful improvements in various spatial audio metrics (like DoA errors) as well as in perceptual studies (MOS scores). The large scale of their new dataset and the multiple comparisons to baselines also strengthen their evidence.

**Essential References Not Discussed:**

I think the references cited are comprehensive.

**Experimental Designs Or Analyses:**

The experiments are well-structured, with a thorough comparison to baselines. The results consistently favor the proposed approach. However, I think more analysis would help clarify a few points:
1. The dual-branch approach is compared against single-branch variants (Section 5 and Table 4), but I would like more clarity on whether combining equirectangular with multiple FOV “cuts” (instead of just one) provides additional improvement.
2. The ratio between non-spatial data and FOA data for pre-training could be further detailed, helping readers understand how general audio tasks complement the specialized FOA domain.

**Methods And Evaluation Criteria:**

The approach focuses on a flow-matching generative model combined with a two-stage training pipeline. They evaluate the generated audio via both objective metrics (e.g., FD, KL divergence, DoA angle errors) and subjective ratings (MOS-SQ for spatial audio quality and MOS-AF for video alignment). These metrics are highly relevant for audio generation tasks. The evaluation on both Sphere360 and an out-of-distribution set (YT360) is also a good indicator of generalization.

**Other Comments Or Suggestions:**

None.

**Other Strengths And Weaknesses:**

I like this paper's motivation and demo, here're some additional weaknesses:
1. I think adding more detail about the variety of ambient scenes or specific events in Sphere360 in the Appendix would help demonstrate coverage.
2. The authors briefly mention the dual-branch system (Section 4.3), but I found the exact architectural interplay between the local and global features a bit missing. A more detailed figure of how these features combine could help.
3. The authors should consider discussing the inference latency of this model, as it is an important factor for AR/VR applications.

**Questions For Authors:**

See the content in the above sections.

**Relation To Broader Scientific Literature:**

This paper advances video-to-audio generation by focusing on FOA audio from 360-degree sources, which has a meaningful place in multi-modal generative modeling.

**Theoretical Claims:**

I did not find major issues with the theoretical discussion -- it is mostly referencing known generative modeling frameworks.

---

> ### Author Rebuttal · Authors · 2025-04-01
>
> Thank you for recognizing our motivation and demo, as well as acknowledging our experimental validations. We hope our response addresses all your concerns and questions.
>
> ## Updated Table 2 and Inference Latency
>
> We sincerely apologize for the typos in Table 2 in the submission. Please check our **Response to Reviewer Xvna** under **Claims that AudioSpace achieves SOTA performance**  As shown in the updated Table 2, **AudioSpace indeed achieves state-of-the-art performance on both in-domain Sphere360 and out-domain YT360-Test.** We also added the number of parameters and inference time into comparison. Although AudioSpace has slightly more parameters than SOTA MMAudio+AS, **it achieves notably faster inference than all baselines.**
>
> ## Combining Equirectangular Projection with Multiple FOV Cuts
>
> We are grateful for your suggestion regarding the dual-branch approach and its comparison with dual-branch variants. Our dual-branch design employs a 120° frontal view to concentrate on prominent sound sources while integrating panoramic features as contextual conditioning to overcome the limitations of visual information beyond the central view. To elucidate the impact of combining equirectangular projection with multiple FOV cuts, we have assessed three innovative FOV-cut strategies to replace the local FOV video: hexadirectional 360° cuts (ERP+6cuts) which capture views from front, back, left, right, up, and down directions, quadrant cuts (ERP+4cuts) which capture views from front, left, right, and back directions, and bipolar cuts (ERP+2cuts) which capture views from front and back directions.
>
> | Dual-Branch Method| FD   | KL   | $\Delta_{abs}\theta$ | $\Delta_{abs}\phi$ | $\Delta_{Angular}$ |
> |--------------------|------|------|----------------------|--------------------|--------------------|
> | ERP+Front          | **88.30** | **1.58** | **1.36** |0.52 | 1.28 |
> | ERP+2FOV Cuts      | 95.84 | 1.77 |  1.37  |  0.55 | 1.30 |
> | ERP+4FOV Cuts      | 92.89 | 1.65 |1.36| **0.51** | 1.27 |
> | ERP+6FOV Cuts      | 90.16 | 1.59 | 1.37 | 0.52 | **1.26**   |
>
> The results show that **our dual-branch strategy still achieves the best audio quality**, reflected in FD and KL, and **achieves similar spatial metrics compared to multiple FOV views**.
>
> ## Ratio between non-spatial data and FOA data for pre-training
>
> The pre-training corpus consists of approximately 2M samples from general non-spatial audio datasets (detailed in Section 5.1) and 100K samples from the specialized Sphere360 FOA dataset, yielding a ratio of 20:1 between non-spatial and spatial audio data. Table 3 shows that coarse-to-fine pre-training substantially improves FD and KL than pre-training with non-spatial data only or FOA data only, confirming that general audio tasks effectively complement the specialized FOA domain.
>
> ## Open-Source Sphere360 Dataset
>
> We fully agree that transparency and reproducibility are critical for the research community.
>
> 1. Open-Source Release: Sphere360 dataset with metadata (including all `youtube_id` identifiers and timestamps) and semi-automated construction pipeline are released at https://anonymous.4open.science/r/Sphere360-CF51/.
>
> 2. Legal-Compliant Data Access
>    Due to YouTube's Terms of Service restrictions, we cannot directly redistribute original videos. Instead, we provide a version-controlled crawling script that reconstructs the raw dataset using the provided `youtube_id` list.
>
> This framework ensures that researchers can fully reconstruct our dataset while complying with content distribution policies. We will refine the documentation based on community feedback to ease the procedure.
>
> ## More Different Audio Events in Appendix
>
> Thank you for the valuable suggestion. We will expand Appendix by incorporating additional examples of diverse audio events to better demonstrate the coverage and diversity of our dataset.
>
> ## More Detailed Figure for Dual-Branch Architecture
>
> Thank you for highlighting this critical architectural detail. **We will revise Figure 10 to explicitly illustrate the dual-branch fusion process** with the following additions:
>
> 1. Local Feature Pathway (FOV-Audio Alignment)
>    * Visual FOV patches are processed through a linear layer to align their dimensionality with audio latents.
>    * The adapted FOV features are directly combined with audio latents via element-wise addition before entering the Diffusion Transformer, ensuring pixel-level spatial correspondence.
>
> 2. Global Context Pathway (360° Scene Guidance)
>    * The full 360° video features are condensed into a global descriptor using max-pooling, capturing holistic scene semantics.
>    * This global descriptor is fused with the diffusion timestep embedding through element-wise addition, providing consistent scene-level conditioning across all transformer layers.

---

> > ### Comment · Reviewer_nf2N · 2025-04-03
> >
> > Thank you for addressing my concerns and including the additional experiments. From my perspective, the paper has made enough contributions, and I will raise my rating to Accept unless Reviewer Xvna identifies further major concerns (the initial concerns could be addressed after reading the rebuttal). Please include all new experiments in the revised version.

---

> > > ### Author Response · Authors · 2025-04-04
> > >
> > > Thank you sincerely for your time, expertise, and constructive engagement throughout the review process. We are deeply grateful for your recognition of our efforts to address the concerns raised earlier, and for your decision to improve the overall recommendation of our paper. Your valuable feedback has been instrumental in significantly enhancing the quality of our work.
> > >
> > > ## Updates for Reproducibility and Quality:
> > > To further facilitate open research, community engagement, and enhance the quality of our paper, we have implemented three key improvements:
> > > 1. **Inference Code Release**
> > > We have open-sourced the inference code via the anonymous repository: https://anonymous.4open.science/r/Audiospace-1348/. Due to the model size (>10GB) and anonymity constraints, we are currently exploring feasible methods to share the pre-trained weights. We will update the repository promptly once resolved. Additionally, we are actively organizing the training code and plan to release it by the end of this month.
> > > 2. **Enhanced Dataset Documentation**
> > > The Sphere360 dataset repository (https://anonymous.4open.science/r/Sphere360-CF51/) has been updated with enhanced documentation, including clearer usage guidelines and dataset structure descriptions.
> > > 3. **Manuscript Revisions**
> > > All suggestions and experimental results discussed during the rebuttal phase will be carefully incorporated into the revised manuscript to ensure a comprehensive presentation of our contributions.
> > >
> > > Should any further clarifications or adjustments be needed, please feel free to share your concerns—we are fully committed to addressing them promptly.

---

### Official Review · Reviewer_MEu5 · 2025-03-12

**Overall Recommendation:** 4

**Summary:**

This paper addresses an interesting problem of generating spatial audio from panoramic videos. They first propose a real-world dataset, Sphere360, for 360 videos and their spatial audios.  They also propose an effective training strategy combining coarse-to-fine pre-training and dual-branch video encoding for spatial-aware generation. The proposed method, AudioSpace, achieves state-of-the-art performance on the Sphere360-Bench

**Claims And Evidence:**

* The high-quality demo videos show great performance in generating spatial audio from the 360 video.

* The quantity results, e.g., Tab.2, demonstrate significant performance improvement of the proposed method.

**Essential References Not Discussed:**

None

**Experimental Designs Or Analyses:**

The experiments and ablation study are well-designed.

I want to see more visual comparisons for the ablation study.

**Methods And Evaluation Criteria:**

The evaluation is reasonable.

For the proposed method:
There are two main stages:  (1) A coarse-to-fine self-supervised flow matching pre-training to alleviate the issue of data scarcity using both unlabeled spatial and non-spatial audio. (2) Fine-tuning the diffusion transformer by efficiently integrating panoramic video representation.

**Other Comments Or Suggestions:**

The motivation and quality of this paper are very good and I would have increased my score if the author had answered my questions adequately at the rebuttal.

**Other Strengths And Weaknesses:**

Weaknesses:
* As the methodology consists of two phases of training, are there relevant experiments analyzing the respective roles of these two phases of training and the relationship between the two phases of training?
* I noticed that the demo videos are all single scenes, what would happen if the given 360 video had scene transitions? And the authors should discuss what the length limit of a 360 video should be.

**Questions For Authors:**

Refer to the "Other Strengths And Weaknesses".

**Relation To Broader Scientific Literature:**

Relate to some research area of CG.

**Theoretical Claims:**

There is no proof for theoretical claims

---

> ### Author Rebuttal · Authors · 2025-04-01
>
> We sincerely appreciate your recognition of the strong motivation and exceptional quality of our paper. We hope our response thoroughly addresses your concerns and questions.
>
> ## Updated Table 2 and Inference Latency
>
> We sincerely apologize for the typos in Table 2 in the submission. Please check our **Response to Reviewer Xvna** under **Claims that AudioSpace achieves SOTA performance.** As shown in the updated Table 2, **AudioSpace indeed achieves state-of-the-art performance on both in-domain Sphere360 and out-domain YT360-Test.** We also added the number of parameters and inference time into comparison. Although AudioSpace has slightly more parameters than MMAudio+AS, **it achieves notably faster inference than all baselines.**
>
> ## Roles of Training Stages
>
> AudioSpace includes a coarse-to-fine self-supervised pre-training stage (**Stage 1**) and a spatial-aware supervised fine-tuning stage (**Stage 2**).
>
> ### 1. Roles of Training Stages
>
> **Stage 1**: Utilizes large-scale unlabeled audio datasets (~2M samples) to build foundational audio understanding capabilities, crucial for 360-degree video-to-spatial audio generation. It focuses on **General Audio Distribution Learning** through masked audio context modeling, enhancing audio feature representation and temporal coherence.
>
> **Stage 2**: Tailors the model for spatial audio generation from 360-degree video. Key aspects of Stage 2 are:
>
> * **Modality Alignment**: Integrates video data for precise spatial audio generation.
> * **Task-Specific Optimization**: Refines the model for the 360V2SA goal, balancing audio quality and spatial accuracy.
>
> ### 2. Validation on 360V2SA Task
>
> | Configuration | FD | KL |
> | --- | --- | --- |
> | Stage 2 only (full data) | 104.57 | 1.83 |
> | Stage 1+2 (full data) | **88.30** | **1.58** |
> | Stage 1+2 (80% data) | 89.86 | 1.60 |
> | Stage 1+2 (60% data) | 93.11 | 1.80 |
> | Stage 1+2 (40% data) | 105.26 | 1.88 |
>
> The table shows that after Stage 1, using 40% of Stage 2 data matches Stage 2 full-data baseline, 60% outperforms it, and 100% substantially exceeds it, in FD and KL: Stage 1 leads to reduced reliance on labeled data and increased robustness.
>
> ### 3. Further Validation on V2A Task
>
> We validate the effects of training stages on traditional V2A tasks on Sphere360. Stage 1 reduces FD by 16.3% relative and KL by 19.2% relative, confirming its effectiveness in improving audio quality.
>
> | Stage | FD | KL |
> | --- | --- | --- |
> | Stage 2 | 41.30 | 2.03 |
> | Stage 1+2 | **34.56** | **1.64** |
>
> The two stages complement each other. Stage 1 builds a strong audio base, while Stage 2 adds modality-specific constraints, preventing overfitting to limited video-audio pairs. The tables highlight the necessity of both stages. Without pre-training, the model struggles with coherent audio, even with video guidance.
>
> ## Video Length Constraints
>
> Our training uses fixed 10-second clips and rotary positional encoding rather than absolute positional embedding for flexible inference lengths. However, the 10-second limit of the video datasets restricts generated audio to under 10 seconds.
>
> The self-supervised pre-training helps with audio coherence, but reliability beyond 10 seconds is uncertain. This is due to potential degradation in modality alignment and feature consistency. Future research is needed to develop extended temporal modeling for long-form audio synthesis.
>
> ## Scene Transition Cases
>
> Thank you for raising this interesting and insightful question.
> Our training and demo data mainly use fixed-camera 360° videos for immersive experiences. To test AudioSpace’s robustness to scene transitions, we used 50 single-scene videos, paired them into 25 groups, and spliced 5-second segments into 10-second clips with abrupt transitions. **We added interactive spatial audio for these clips on the demo page - [Scene Transition Cases]**. Results show that in some examples, our model can generate transitions that are natural and seamless. It even integrates the content from both segments into a coherent whole—for instance, when the two segments showcase performances by different musical instruments, the synthesis can merge them into a balanced duet at times. However, because our training data predominantly consists of single-scene videos, the model occasionally emphasizes one segment over the other, sometimes overlooking the acoustic nuances of the less dominant scene.
>
> ## More Visual Comparisons for Ablation Studies
> Due to the time limit of the first rebuttal phase, we will add more visual comparisons for the ablation study in the final response.

---

> > ### Comment · Reviewer_MEu5 · 2025-04-03
> >
> > Thanks for the authors' responses, and I will keep my acceptance.

---

> > > ### Author Response · Authors · 2025-04-04
> > >
> > > Thank you sincerely for your time, expertise, and constructive engagement throughout the review process. We deeply appreciate your recognition of our work, as well as your valuable feedback that has significantly improved the quality of this work.
> > >
> > > ## More Visual Ablation Comparisons
> > >
> > > As acknowledged during the initial review phase, time constraints limited our inclusion of detailed ablation visualizations. In direct response to your feedback, we have now comprehensively updated the **[Demo Page - Ablation Study]** with systematic comparisons. These visualizations explicitly demonstrate the contributions of individual components in our framework, aligning with your suggestions for methodological transparency.
> > >
> > > ## Updates for Reproducibility and Quality:
> > > To further facilitate open research, community engagement, and enhance the quality of our paper, we have implemented three key improvements:
> > > 1. **Inference Code Release**
> > > We have open-sourced the inference code via the anonymous repository: https://anonymous.4open.science/r/Audiospace-1348/. Due to the model size (>10GB) and anonymity constraints, we are currently exploring feasible methods to share the pre-trained weights. We will update the repository promptly once resolved. Additionally, we are actively organizing the training code and plan to release it by the end of this month.
> > > 2. **Enhanced Dataset Documentation**
> > > The Sphere360 dataset repository (https://anonymous.4open.science/r/Sphere360-CF51/) has been updated with enhanced documentation, including clearer usage guidelines and dataset structure descriptions.
> > > 3. **Manuscript Revisions**
> > > All suggestions and experimental results discussed during the rebuttal phase will be carefully incorporated into the revised manuscript to ensure a comprehensive presentation of our contributions.
> > >
> > > Should any further clarifications or adjustments be needed, please feel free to share your concerns—we are fully committed to addressing them promptly.

---

### Official Review · Reviewer_Xvna · 2025-03-13

**Overall Recommendation:** 3

**Summary:**

This paper proposes the task of generating spatial audio from 360-degree videos. To support this task, the authors construct a dataset named Sphere360, comprising curated real-world 360-degree videos collected from YouTube. Leveraging this dataset, the authors introduce the AudioSpace model, which employs self-supervised pretraining with both spatial and non-spatial datasets, along with a dual-branch architecture utilizing panoramic and field-of-view (FOV) video inputs. Given the preliminary nature of this task, several baseline models are established for comparison. Experimental results indicate that AudioSpace achieves promising outcomes in both objective and subjective evaluations.

**Claims And Evidence:**

- The claim that AudioSpace achieves state-of-the-art (SOTA) performance is questionable. The authors frequently emphasize their final model's results in bold throughout Tables 2, 3, and 4, even in cases of tied scores or superior performance by other models. This could mislead readers into perceiving AudioSpace as consistently superior across all metrics.

- The introduced 360V2SA dataset represents a valuable contribution to the community, facilitating further research into 360-degree video-guided spatial audio generation.

**Essential References Not Discussed:**

N/A

**Experimental Designs Or Analyses:**

- The comparison setup may not be entirely fair, as existing methods are trained exclusively on Sphere360, whereas AudioSpace undergoes initial training on FreeSound/AudioSet/VGGSound before Sphere360. Conducting comparisons with identical training datasets across all methods would better highlight the effectiveness of the proposed model. Additionally, the authors note in Table 3 that coarse training significantly contributes to performance; thus, applying similar training strategies to existing methods would clarify the specific impact of the proposed model design.

- In Table 2, specifically for $\Delta_{abs}\phi$ in YT360-Test, AudioSpace is incorrectly marked in bold despite performing worse than ViSAGe. Similar inconsistencies appear in Table 3, where numbers ($\Delta_{angular}$) are incorrectly highlighted or tied results ($\Delta_{abs}\theta$) are unnecessarily emphasized for AudioSpace. Such inconsistencies may create misleading interpretations, suggesting the authors intentionally emphasize favorable results. Similar issues appear in Table 4.

**Methods And Evaluation Criteria:**

- In Section 3 (Data Cleaning), the authors state that videos with an audio-visual similarity score below 1 are discarded. Clarification is needed regarding this threshold. Specifically, what is the range of audio-visual similarity scores? If this score refers to cosine similarity, the threshold value provided may be incorrect or misleading.

- During 360-degree video-guided fine-tuning, authors mention max-pooling 360 features to serve as a global condition. However, extracting only a single vector from the image encoder might not sufficiently represent motion information (e.g., a car moving from right to left). Additional explanation or justification of this method is necessary.

- Are the outputs depicted in Figure 2(b) fed directly into the decoder of the spatial VAE? The inference of each component requires clearer explanation.

- The selection method for the field of view (FOV) input in the visual encoder is not explained. Additionally, how should the FOV be determined during test time? Clarification on FOV selection criteria is necessary.

**Other Comments Or Suggestions:**

- Line 158 references Table 8, which does not contain information about audio event distribution as suggested.

**Other Strengths And Weaknesses:**

See other sections.

**Questions For Authors:**

N/A

**Relation To Broader Scientific Literature:**

No explicit relation to broader scientific literature is identified.

**Theoretical Claims:**

No theoretical claims are presented.

---

> ### Author Rebuttal · Authors · 2025-04-01
>
> We sincerely appreciate your constructive feedback and detailed suggestions. We hope our response below fully resolves your concerns and questions.
>
> ## Claims that AudioSpace achieves SOTA performance
> We sincerely apologize for the two typos in Table 2 main results and typos and boldfacing errors in Table 3 and Table 4. These were completely unintentional oversights on our part. **As shown in the updated Table 2 below, AudioSpace indeed achieves state-of-the-art performance on both in-domain Sphere360 and out-domain YT360-Test.**
> ### 1. Open-Source Commitment
> Codebase and model weights will be open-sourced by April 2025. The Sphere360 dataset is already released at https://anonymous.4open.science/r/Sphere360-CF51/.
> ### 2.Updated Table 2 (Bold=Best, Highlight fixed numbers):
> Fixed typos in YT360-Test results. AudioSpace achieves **best objective and subjective results and fastest inference speed**:
>
> |Model|FD|KL|$\Delta_{abs}\theta$|$\Delta_{abs}\phi$|$\Delta_{Angular}$|Params|Infer Time(s)|TFlops|
> |---|---|---|---|---|---|---|---|---|
> |Diff-Foley+AS|361.65|2.22|/|/|/|0.94B|2.40|9.11|
> |MMAudio+AS|190.40|`1.71`|/|/|/|1.03B|3.01|7.18|
> |ViSAGe(FOV)|199.09|1.86|2.21|0.88|1.99|0.36B|22.37|0.68|
> |ViSAGe(360)|225.52|1.95|2.18|0.86|1.98|0.36B|22.37|0.68|
> |AudioSpace|**92.57**|**1.64**|**1.27**|**`0.53`**|**1.27**|1.22B|**0.92**|21.15|
>
> ### 3. Extended V2A Validation:
> AudioSpace also excels in video-to-audio generation. Please see **Response to Reviewer cQFE** for results.
>
> ### 4. Fixed Table 3 and Table 4:
>   AudioSpace results (row1 in both tables) are correct and best in FD, KL, and most spatial metrics. Fixed Tables 3 and 4 are on the demo page. **Coarse-to-fine pre-training and dual-branch design are superior to alternatives**.
>
> ## Fairness of Experimental Comparisons
>
> ### 1.Fair Comparison Protocol
> All baselines in Table 2 follow their original pre-training and are fine-tuned on Sphere360 with the same setup.
>
> |Model|Training Data (Type/Scale)|Pretraining Strategy|
> |---|---|---|
> |ViSAGe|VGGSound/YT-360 (~300k pairs)|Video-guided init.|
> |MMAudio|VGGSound/WavCaps/AudioCaps (~1.1M pairs)|Text+video init.|
> |AudioSpace|FreeSound/AudioSet/VGGSound/Sphere360 (~2M audio samples)|**Audio-centric pretraining**|
>
> **Uniform pretraining is unsuitable as baselines need multimodal pairs (text/video+audio)**, while AudioSpace focuses on **audio diversity** through domain-specific pretraining.
>
> ### 2. Architectural Superiority: Table 3 **w/o PT** (trained from scratch on Sphere360) and Table 2 show structural advantages: **Without pre-training, AudioSpace surpasses all pretrained-finetuned baselines on Sphere360 in all objective metrics.**
>
> ## Threshold for audio-visual similarity score for data cleaning
> The threshold of 1 corresponds to the original cosine similarity score scaled by 10 for easier processing, which does not impact comparative results or conclusions. The threshold was chosen based on video-audio alignment control and data analysis.
>
> ## 360˚ Video Representation
> We appreciate the question about motion representation in ERP-formatted videos. Our dual-branch design addresses ERP limitations as follows:
>
> ### 1. Geometric Distortion in ERP
> ERP causes planar distortions and boundary discontinuities, conflicting with CLIP encoders’ planar continuity assumptions, leading to feature misalignment. Table 4 shows single-branch ERP-only model suffers from severe feature distortion (FD: 97.83 vs. our 88.30) and semantic inconsistency (KL: 1.87 vs. our 1.58), indicating naive planar encoding fails in spherical contexts.
>
> ### 2. Dual-Branch Design with Frontal Perspective
> ERP distorts global spatial relationships, but motion trajectories remain coherent within FOV. Our temporal modeling FoV branch isolates motion modeling to distortion-free frontal regions, ensuring stable motion encoding, as shown by Y-channel intensity variations in Fig.13.
>
> ### 3. Limitations and Future Directions
> Objects entirely outside the frontal viewport may challenge our current approach, though they represent <5% of training data in typical 360° videos. Future work will explore geometric-aware feature rectification to address ERP distortions at polar regions.
>
> ## Clarification of Inference
> During inference, the frontal-view segment of the panoramic video is used for local conditioning, and the full panoramic video for global conditioning. Both features guide the diffusion transformer's ODE solver (Euler method, CFG scale=5). The audio patents are decoded by the pre-trained spatial VAE decoder to produce the final spatial audio outputs.
>
> ## FOV selection criteria
> During training and testing, we use the frontal-view large FOV (120°) as the default FOV video. This choice aligns with real-world conditions where front-facing viewpoints capture main sound sources.
>
> ## Typos
> Thank you for your reminder. Line 158 should refer to Figure 8, not Table 8, for the audio event distribution. We will make all the fixes in the revised paper.

---

> > ### Comment · Reviewer_Xvna · 2025-04-06
> >
> > Thank you for the detailed response. Most of my concerns are resolved and will update by ratings to weak accept.

---

> > > ### Author Response · Authors · 2025-04-06
> > >
> > > Thank you sincerely for your time, expertise, and constructive engagement throughout the review process. We are deeply grateful for your recognition of our efforts to address the concerns raised earlier, and for your decision to improve the overall recommendation of our paper. Your valuable feedback has been instrumental in significantly enhancing the quality of our work.
> > >
> > > ## Updates for Reproducibility and Quality:
> > > To further facilitate open research, community engagement, and enhance the quality of our paper, we have implemented three key improvements:
> > > 1. **Inference Code Release**
> > > We have open-sourced the inference code via the anonymous repository: https://anonymous.4open.science/r/Audiospace-1348/. Due to the model size (>10GB) and anonymity constraints, we are currently exploring feasible methods to share the pre-trained weights. We will update the repository promptly once resolved. Additionally, we are actively organizing the training code and plan to release it by the end of this month.
> > > 2. **Enhanced Dataset Documentation**
> > > The Sphere360 dataset repository (https://anonymous.4open.science/r/Sphere360-CF51/) has been updated with enhanced documentation, including clearer usage guidelines and dataset structure descriptions.
> > > 3. **Manuscript Revisions**
> > > All suggestions and experimental results discussed during the rebuttal phase will be carefully incorporated into the revised manuscript to ensure a comprehensive presentation of our contributions.
> > >
> > > Should any further clarifications or adjustments be needed, please feel free to share your concerns—we are fully committed to addressing them promptly.

---

### Decision · Program_Chairs · 2025-05-01

**Decision:**

Accept (poster)

**Comment:**

The paper received consistently positive feedback following the authors’ rebuttal. Reviewers highlighted the paper's experimental validation, and the quality of the results with the high-quality demo (cQFE, nf2N, MEu5). While there were initial concerns about the comparison setup and presentation errors in the table, these were effectively addressed in the rebuttal, leading to stronger overall reviews.

After carefully reviewing the paper, the reviews, and the rebuttal, The AC concurs with the reviewers' positive consensus and recommends acceptance of this paper.

For the camera-ready version, the authors should integrate all discussions from the rebuttal into the main paper and supplementary materials. Specifically, the authors should implement the following changes:
1. Manuscript Revisions: typos and boldfacing errors in the tables (nf2N, Xvna). The usage of terminology and more ablations on the dual-branch design and validations (cQFE). More different audio events in appendix (nf2N).
2. Inference Code Release
3. Enhanced Dataset Documentation